# Building your own mountain: The effects, limits, and drawbacks of cold-water coral ecosystem engineering

Anna-Selma van der Kaaden[1,2], Sandra R. Maier[1,3], Siluo Chen[1,2], Laurence H. De Clippele[4], Evert de Froe[5], Theo Gerkema[1], Johan van de Koppel[1,6], Furu Mienis[5], Christian Mohn[7], Max Rietkerk[2], Karline Soetaert[1], Dick van Oevelen[1]

[1]NIOZ Royal Netherlands Institute for Sea Research, Department of Estuarine and Delta Systems, PO Box 140, 4400 AC Yerseke, The Netherlands
[2]Copernicus Institute for Sustainable Development, Department of Environmental Sciences, Utrecht University, The Netherlands
[3]Greenland Climate Research Centre, Greenland Institute of Natural Resources, Nuuk, Greenland
[4]University of Glasgow, School of Biodiversity, One Health, and Veterinary Medicine
[5]Department of Ocean Systems, NIOZ Royal Netherlands Institute for Sea Research, The Netherlands
[6]Conservation Ecology Group, Groningen Institute for Evolutionary Life Sciences, University of Groningen, the Netherlands
[7]Deparament of Ecoscience, Aarhus University, Roskilde, Denmark

*Correspondence to*: Anna van der Kaaden (annavanderkaaden@gmail.com)

**Abstract.** Framework-forming cold-water corals are ecosystem engineers that build mounds in the deep sea that can be up to several hundred meters high. The effect of the presence of cold-water coral mounds on their surrounding is typically difficult to separate from environmental factors that are not affected by the mounds. We investigated the environmental control on and the importance of ecosystem engineering for cold-water coral reefs, using annotated video transects data, spatial variables (MEMs) and hydrodynamic model output in a redundancy analysis and with variance partitioning. Using available hydrodynamic simulations with cold-water coral mounds and simulations where the mounds were artificially removed, we investigated the effect of coral mound ecosystem engineering on the spatial configuration of reef habitat and discriminated which environmental factors are and which are not affected by the mounds.

We find that downward velocities in winter, related to non-engineered environmental factors, e.g., deep winter mixing and dense water cascading, cause substantial differences in reef cover at the broadest spatial scale (20-30 km). Such hydrodynamic processes that stimulate the food supply towards the corals in winter seem more important for the reefs than cold-water coral mound engineering or than similar hydrodynamic processes in summer. While the ecosystem engineering effect of cold-water corals is frequently discussed, our results highlight also the importance of non-engineered environmental processes.

We further find that, due to the interaction between the coral mound and the water flow, different hydrodynamic zones are found on coral mounds that likely determine the typical benthic zonation of coral rubble at the mound foot, dead coral framework on the mound flanks, and living corals near the summit. Moreover, we suggest that a so-called massenerhebung effect (well-known for terrestrial mountains) exists, meaning that benthic zonation depends on the location on the mound rather than on the height above the seafloor or water depth. Our finding that ecosystem engineering determines the configuration of

benthic habitats on cold-water coral mounds, implies that cold-water corals cannot grow at deeper depths on the mounds to avoid the adverse effects of climate change.

## 1. Introduction

Framework-forming cold-water corals are benthic suspension feeders that rely on particulate organic matter originating from the sunlit ocean surface (van Oevelen et al., 2018; Van Engeland et al., 2019; Carlier et al., 2009). They build habitats of live and dead reef framework for many species and have a high biodiversity and productivity as compared to the surrounding deep-sea environment (van Oevelen et al., 2009; Cathalot et al., 2015; Bongiorni et al., 2010). Many of the associated reef-fauna are also suspension feeders that require similar food advection by currents as the framework-forming corals themselves (Maier et al., 2021). With their structurally complex framework, cold-water corals engineer their environment, increasing the food supply towards themselves and their associated fauna (van der Kaaden et al., 2020; Hennige et al., 2021; Bartzke et al., 2021; Mienis et al., 2019). These important deep-sea ecosystems are expected to be severely impacted by ocean warming and acidification (Hennige et al., 2020; Morato et al., 2020; Ragnarsson et al., 2016; Sweetman et al., 2017) and understanding what drives cold-water coral reef growth at different spatial scales provides insights into their expected response to contemporary global changes.

When the positive feedbacks of an organism on its own growth are strong enough, the ecosystem engineer can locally create its own optimal environment and exist in suboptimal ambient environmental conditions where it would not survive otherwise (Jones et al., 1994; Crooks, 2002; Hastings et al., 2007). Recently, reef-forming cold-water corals were identified as a 'self-organized' system, meaning that the corals enhance their resource intake by optimizing their spatial configuration on the seafloor (van der Kaaden et al., 2023). By adapting their spatial configuration in the ecosystem, ecosystem engineers can respond gradually to environmental changes, providing resilience in the face of relatively rapid global change (Bastiaansen et al., 2020; Rietkerk et al., 2021).

Framework-forming cold-water corals also engineer their environment at much larger spatial scales, because the reefs can develop into mounds, where coral growth and sediment infill on the reefs is higher than in the off-reef area (Wang et al., 2021; van der Land et al., 2014). These mounds can be hundreds of meters high, enabling corals to feed higher in the water column where particulate organic matter quality and quantity is higher (Snelgrove et al., 2017; Nakatsuka et al., 1997). Moreover, the interaction of mounds with (tidal) currents also increases the food supply towards the coral reefs (van der Kaaden et al., 2021). Internal (tidal) waves, created around the mounds (van Haren et al., 2014; Cyr et al., 2016; Davies et al., 2009) increase the horizontal and vertical food flux towards the reefs (Duineveld et al., 2012; Froe et al., 2022; Soetaert et al., 2016).

Soetaert et al. (2016) simulated the flow of particulate organic matter to cold-water coral reefs in the Logachev cold-water coral mound province with a coupled 3D model. They found that the flux of particulate organic matter is concentrated towards the corals living on the mounds due to the physical presence of the mounds. Recent field observations indeed indicate the existence of this food transport pathway (de Froe et al., 2022). Van der Kaaden et al. (2021) further investigated the interaction

of coral mounds and hydrodynamics for various mound sizes. For this, the bathymetry was modified to mimic different mound sizes, and used in 3D hydrodynamic simulations. They found that, like coral reefs, coral mounds exert several spatial feedbacks through ecosystem engineering of bottom currents and vertical velocities. So, the ecosystem engineering effect of framework-forming cold-water corals is well-established at both the scale of reefs and of coral mounds. However, how the effect of cold-water coral mounds on the hydrodynamics (i.e., 'coral mound engineering') affects the spatial configuration of cold-water coral reef habitats, and their associated fauna, remains unclear.

Furthermore, separating those environmental factors that are affected by coral mound engineering and those environmental factors that are not affected by cold-water corals is seemingly impossible since one cannot simply remove all coral mounds and compare measurements of the environment with and without (i.e., smoothed topography) mounds. In this study, we make use of the unique situation at the Logachev cold-water coral mound province for which hydrodynamic model simulations with and without coral mounds are available (van der Kaaden et al., 2021). Specifically, we combined this hydrodynamic model output with annotated video transects from Maier et al. (2021) and De Clippele et al. (2021), and variables identifying spatial scale (Moran Eigenvector Maps) to identify the main environmental drivers of reef cover in a redundancy analysis (RDA). Variance partitioning showed how much of the spatial structure of reef cover is related to the different hydrodynamic variables and by comparing hydrodynamic simulations with and without coral mounds, we investigated which hydrodynamic variables are caused by feedbacks between the mounds and the hydrodynamics. This combined approach allowed us to investigate the environmental control on reef cover at different spatial scales and discriminate the effects of coral mound engineering and non-engineered environmental factors.

## 2. Methods

### 2.1 Site description

The Logachev cold-water coral mound province (Fig. 1) is located west of Ireland on the SE Rockall Bank margin (NE Atlantic) (de Haas et al., 2009; Kenyon et al., 2003; Mienis et al., 2006). Cold-water coral mound clusters occur between 600 m and 1,100 m depth, usually in the Wyville Thomson Overflow Water (WTOW) current that flows in SW direction (Schulz et al., 2020; Johnson et al., 2010). The mounds are situated at the location of increased internal-tide generation as compared to the rest of the SE Rockall Bank margin (van der Kaaden et al., 2021). Mound summits are situated around the depth of the permanent pycnocline (White and Dorschel, 2010). A bottom-trapped internal tidal wave at diurnal frequency travels past the Rockall Bank margin, causing internal waves with amplitudes of over 120 m near the mounds (van Haren et al., 2014).

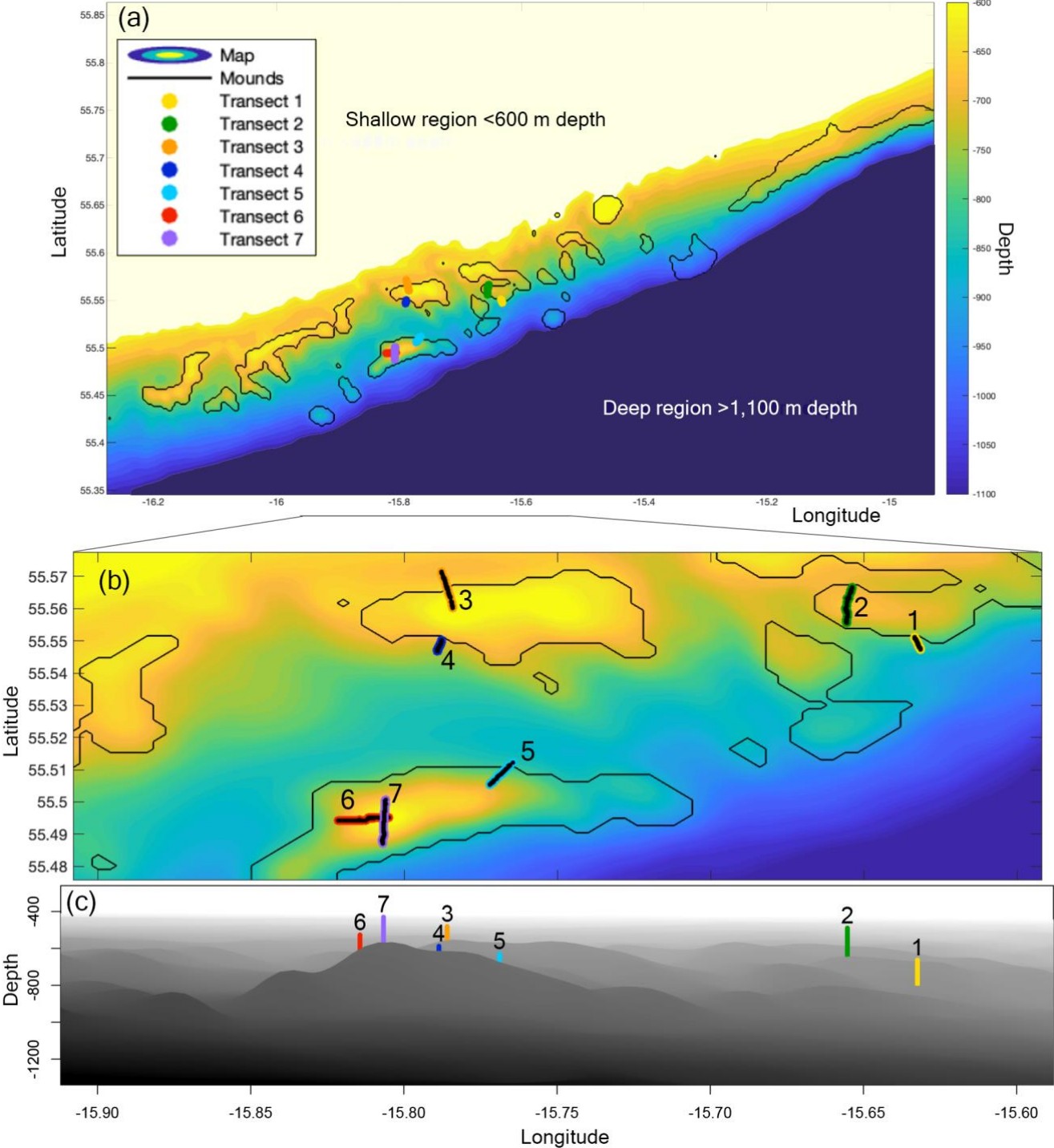

**Figure 1. (a) Map of the Logachev cold-water coral mound province, as used in the hydrodynamic simulations, depicting the depth (m) in colour. The mounds, as selected by connected-component labelling, are encircled by black lines. The seven transects are plotted in colour. (b) A zoom in on the Logachev cold-water coral mound province on the seven transects. The transects are**

95

numbered and highlighted in black for clarity. (c) A side view of the zoom in on Logachev coral mound province with the centre of the transects indicated by the coloured line. Bathymetric data are derived from the Irish National Seabed Survey (INSS) at 250 m resolution.

## 2.2 Response and explanatory variables

Redundancy Analysis (RDA) identifies the main trends in response variables (i.e., benthic cover) along continuous axes and how these trends correlate to explanatory variables (i.e., depth, hydrodynamic and spatial variables) (Borcard et al., 2011). Below, we first describe how response and explanatory variables were obtained from the available data (Table 1) and then detail the RDA approach.

### 2.2.1 Video frame extraction and annotation (response variables)

Video transects 1 to 6 were recorded with the Remotely Operated Vehicle (ROV) *Holland-1* between 600 m and 850 m depth, during the Changing Oceans Expedition 2012 Cruise 073 with the RRS James Cook. The ROV was equipped with a Reson 7125 multibeam system and a downward-facing HD camera for continuous recording and two lasers for size reference (Roberts and shipboard party, 2013). From these videos, 1.196 frames were manually annotated by De Clippele et al. (2021) for live reef-forming corals, dead coral framework, coral rubble, rocks, and sediment, following the annotation protocol by van der Kaaden and De Clippele (2021).

Video transect 7 was recorded during the *RV Pelagia* cruise 64PE360 in October 2012 between 600 m and 700 m depth with a towed camera frame from the NIOZ (Royal Netherlands Institute for Sea Research) and two lasers for size reference. A total of 187 frames were manually annotated by Maier et al. (2021) for live reef-forming corals, less-degraded dead framework, more-degraded dead framework, coral rubble, rocks, sediment, sponges and other macrofauna, following the same annotation protocol (van der Kaaden and De Clippele, 2021). We combined the two datasets for our analysis and used the benthic cover (% of the annotated frames) of live coral, dead coral, rubble, rocks, and sediment as response variables in the RDA.

### 2.2.2 Bathymetry and hydrodynamic variables (explanatory variables)

Bathymetric and hydrodynamic variables were obtained from recently published 3D hydrostatic Regional Ocean Modelling System (ROMS) simulations for the study area (van der Kaaden et al., 2021). The model uses 3D free-surface, hydrostatic equations with grid refinement on a staggered Arakawa C-grid in the horizontal and topography-following coordinates in the vertical (Shchepetkin and McWilliams, 2005). The model covered an area of 85 km x 58 km with a horizontal resolution of 250 m, nested into a model grid of 190 km x 188 km with a resolution of 750 m. In the vertical, the model contained 32 levels with enhanced resolution near the surface and bottom. Tidal forcing (TPXO7) and realistic water (WOA) and atmospheric (COADS) forcings were applied at the boundaries of the lower-resolution model domain. The model was run with climatological forcing and model output was obtained every 6 hours for two runs of 30 days in February and August. A detailed description of the model and selected variables is given in van der Kaaden et al. (2021) and (2022).

We included 30-day averages of the following hydrodynamic variables because they are thought to be related to the coral food supply: horizontal current speed at the bottom layer of the model ("Speed"), upward and downward velocities between 250-300 m depth ("$W_{up}$" and "$W_{down}$"), and depth-integrated energy conversion rate ("EC") to the diurnal (K1) internal tide that is trapped at the Rockall Trough margin (van der Kaaden et al., 2021). The energy conversion rate is a measure of internal tidal wave activity (Gerkema et al., 2004). February and August bottom current speeds and energy conversion rates correlated strongly (Spearman correlation coefficient $r_S=0.81$ and $r_S=0.95$ resp.) which is undesirable in an RDA, so we used the average of the two months (i.e., 60-day average). For each annotated frame, values of all described variables were taken from the model grid cell closest to the position of the respective annotated frames (based on longitude and latitude coordinates).

### 2.2.3 Spatial variables from Moran's Eigenvector Maps (explanatory variables)

To investigate the spatial configuration of reef cover and environmental variables at multiple spatial scales, we used Moran's Eigenvector Maps (MEMs). In the MEM-approach, the study area is divided into groups (i.e., a map with 1 group, 2 groups, 3 groups, etc.) and each annotated video frame is assigned a value that represents its proximity to the centre of the group in which it is located (Borcard et al., 2011). Moran's Eigenvector Map 1 thus represents variations on the broadest spatial scale of the entire study area (~25 km) and the highest MEMs (i.e., when the study area is divided in the highest number of groups) describe variation at the finest spatial scale of the difference between individual annotated video frames (~4 m). MEMs merely describe a certain spatial scale and thus do not exert any control on CWC growth, their specific values are therefore meaningless with respect to CWC growth.

Moran's Eigenvector Maps were created from a distance-based connectivity matrix with the *adespatial* package in R (Dray et al., 2022). The connectivity matrix was calculated from the transformed spherical longitude and latitude coordinates using a minimum distance of 0 and the maximum distance at which all the datapoints were connected. This resulted in 15 significant (Moran's adjusted $p<0.05$) MEMs with positive eigenvalues. At the broadest spatial scale (signalling differences between transects) datapoints were classified according to their east-to-west position (MEM 1) or according to their north-to-south position (MEM 2). At finer scales (signalling differences within transects) datapoints were classified by MEMs according to their position along a mound flank (MEM 3, 4, and 5), or according to regular intervals along transects (MEM 6 to MEM 15). So, MEM 1 and MEM 2 describe the between-transect spatial scale and higher MEMs describe the within-transect spatial scale. Data points on the same transect will thus have very similar values for MEM 1 and 2 but can have very different values for MEM 15.

### 2.3 RDA and variance partitioning

We explored our data following the protocol by Zuur et al. (2010) to ensure compliance to the assumptions of the RDA and variance partitioning. We applied a Hellinger transformation to the benthic cover data to minimize differences in standard deviations and applied scaling during RDA and variance partitioning to homogenise the variable variances. With two variance partitioning analyses, we calculated the percentage of variance in benthic cover that was explained by 1) hydrodynamic

variables, depth, and spatial variables and 2) environmental variables (hydrodynamic variables and depth), between-transect spatial variables, and within-transect spatial variables. The latter analysis provides information on whether benthic cover varies most between transects or within transects.

For parsimonious analyses and to avoid high collinearity, we selected the explanatory variables to be used in RDA and variance partitioning using forward selection with a significance level of 0.05 and the extra stopping criterium based on the adjusted coefficient of multiple determination ($R^2_{adj}$) of the global model (Blanchet et al., 2008). In this way, we selected the variables in the explanatory datasets separately, after verifying that the RDA with variables from the respective dataset was significant (Blanchet et al., 2008). We included depth as a separate variable in the final set of explanatory variables. Forward selection was done in R with the *adespatial* package (Dray et al., 2022), collinearity was tested by calculating Variation Inflation Factors (VIFs) with the *vegan* package, and RDA and variance partitioning were also performed with the *vegan* package (Oksanen et al., 2022).

**Table 1. Response and explanatory variables as used in RDA and variance partitioning, with their units and range (min, mean, max). Vertical velocities are 30-day averages of February or August, and bottom current speed and energy conversion rates are 60-day averages of February+August. The range of values was calculated based on the variables as used in RDA and variance partitioning.**

| | Set | Name | Range: min | mean | max |
|---|---|---|---|---|---|
| **Response variables** | **Benthic cover** | Live coral cover ("Live") | 0 | 3.9 | 61.6 % |
| | | Dead coral cover ("Dead") | 0 | 30.7 | 100 % |
| | | Coral rubble cover ("Rubble") | 0 | 37.6 | 100 % |
| | | Sediment cover ("Sediment") | 0 | 21.1 | 100 % |
| | | Cover of rocks ("Rocks") | 0 | 2.4 | 98.7 % |
| **Explanatory variables** | **Hydrodynamic variables** | Bottom current speed ("Speed") | 0.18 | 0.27 | 0.34 m s$^{-1}$ |
| | | Upward velocities in February ("$W_{up}$ Feb") | 0.001 | 0.002 | 0.003 m s$^{-1}$ |
| | | Upward velocities in August ("$W_{up}$ Aug") | 0.003 | 0.008 | 0.015 m s$^{-1}$ |
| | | Downward velocities in February ("$W_{down}$ Feb") | 0.005 | 0.011 | 0.020 m s$^{-1}$ |
| | | Downward velocities in August ("$W_{down}$ Aug") | 0.002 | 0.005 | 0.011 m s$^{-1}$ |
| | | Energy conversion to the internal tide ("EC") | -0.13 | 0.45 | 0.98 W m$^{-2}$ |
| | **Depth** | Depth | 528 | 682 | 877 m |
| | **Spatial variables** | Between-transect Moran's Eigenvector Maps MEM 1 | -0.7 | 0.0 | 1.6 |
| | | MEM 2 | -1.5 | 0.0 | 2.7 |

| | | | | |
|---|---|---|---|---|
| **Within-transect Moran's Eigenvector Maps** | | | | |
| | **MEM 3** | **-1.9** | **0.0** | **3.1** |
| | **MEM 4** | **-2.9** | **0.0** | **3.0** |
| | **MEM 5** | **-3.3** | **0.0** | **3.1** |
| | **MEM 6** | **-3.2** | **0.0** | **3.5** |
| | **MEM 7** | **-3.9** | **0.0** | **3.5** |
| | **MEM 8** | **-2.9** | **0.0** | **4.1** |
| | **MEM 9** | **-3.7** | **0.0** | **4.3** |
| | **MEM 11** | **-7.7** | **0.0** | **4.4** |
| | **MEM 13** | **-5.1** | **0.0** | **5.0** |
| | **MEM 14** | **-5.5** | **0.0** | **3.6** |
| | **MEM 15** | **-3.9** | **0.0** | **4.9** |

## 2.4 Hydrodynamic simulations with and without coral mounds (coral mound engineering effect)

Some hydrodynamic (i.e., part of the explanatory) variables included in the analyses might be affected by the presence of cold-water coral mounds (i.e., 'coral mound engineering effect'). We try to discriminate the environmental factors that are affected by coral mound engineering and environmental factors that are not affected by cold-water corals by comparing hydrodynamic model simulations with and without cold-water coral mounds from van der Kaaden et al. (2021). Van der Kaaden et al. (2021) mimicked a situation without cold-water coral mounds by smoothing the bathymetry and simulated the hydrodynamics on the smoothed and unmodified bathymetry. We calculated the coral mound engineering effect by subtracting horizontal bottom current speed, upward velocities in February, upward velocities in August, downward velocities in February, downward velocities in August, and energy conversion rates (Table 2) of the simulations with smoothed bathymetry (i.e., without coral mounds) from simulations with unmodified bathymetry (i.e., with coral mounds).

We define a coral mound engineering effect as a positive or negative feedback, meaning that the magnitude of the hydrodynamic variables increases or decreases resp. with increasing coral mound height. To investigate which hydrodynamic variables are influenced by the size of a coral mound we calculated Spearman rho correlation coefficients between the mean absolute effect of mound presence and mound height. A significant correlation thus indicates a significant positive or negative effect of coral mound engineering on the hydrodynamic variable.

Mound height was defined using the model topography from van der Kaaden et al. (2021) as the maximum difference in topography at the location of the mounds, where we determined the location of mounds using connected component labelling in Matlab (Fig. 1). Mound height is therefore the maximum topographic height corrected for the underlying continental slope.

**Table 2. Hydrodynamic variables used to visualize the effect of cold-water coral mounds on the hydrodynamics and to calculate Spearman rho correlation coefficients. The values display the minimum, mean, and maximum values of the hydrodynamic variables at the locations of cold-water coral mounds (Fig. 1) for the unmodified bathymetry (with mounds, left) and for the smoothed bathymetry (without mounds, right). Vertical velocities are 30-day averages of February or August, and bottom current speed and energy conversion rates are 60-day averages of February+August.**

| | With mounds (unmodified bathymetry) | | | Without mounds (smoothed bathymetry) | | |
|---|---|---|---|---|---|---|
| | min | mean | max | min | mean | max |
| **Bottom current speed ("Speed")** | 0.12 | 0.22 | 0.42 m s$^{-1}$ | 0.08 | 0.22 | 0.37 m s$^{-1}$ |
| **Upward velocities in February ("$W_{up}$ Feb")** | 0.002 | 0.007 | 0.029 m s$^{-1}$ | 0.001 | 0.002 | 0.023 m s$^{-1}$ |
| **Upward velocities in August ("$W_{up}$ Aug")** | 0 | 0.004 | 0.018 m s$^{-1}$ | 0 | 0.001 | 0.005 m s$^{-1}$ |
| **Downward velocities in February ("$W_{down}$ Feb")** | 0.001 | 0.007 | 0.022 m s$^{-1}$ | 0.001 | 0.003 | 0.021 m s$^{-1}$ |
| **Downward velocities in August ("$W_{down}$ Aug")** | 0 | 0.003 | 0.013 m s$^{-1}$ | 0 | 0.001 | 0.003 m s$^{-1}$ |
| **Energy conversion to the internal tide ("EC")** | -0.65 | 0.08 | 1.25 W m$^{-2}$ | -0.04 | 0.04 | 0.23 W m$^{-2}$ |

## 3. Results

### 3.1 General transect observations

There is variability in benthic cover within- and between transects (Fig. 2), but clear trends are not obvious. For instance, transect 3, located on the lower southern flank of the northernmost mound, and transect 6, located at the western flank of the highest mound (Fig. 1), have a relatively low live and dead coral cover overall. Transect 7, across the highest mound in the area, shows a relatively high dead coral cover and low sediment and rubble cover overall.

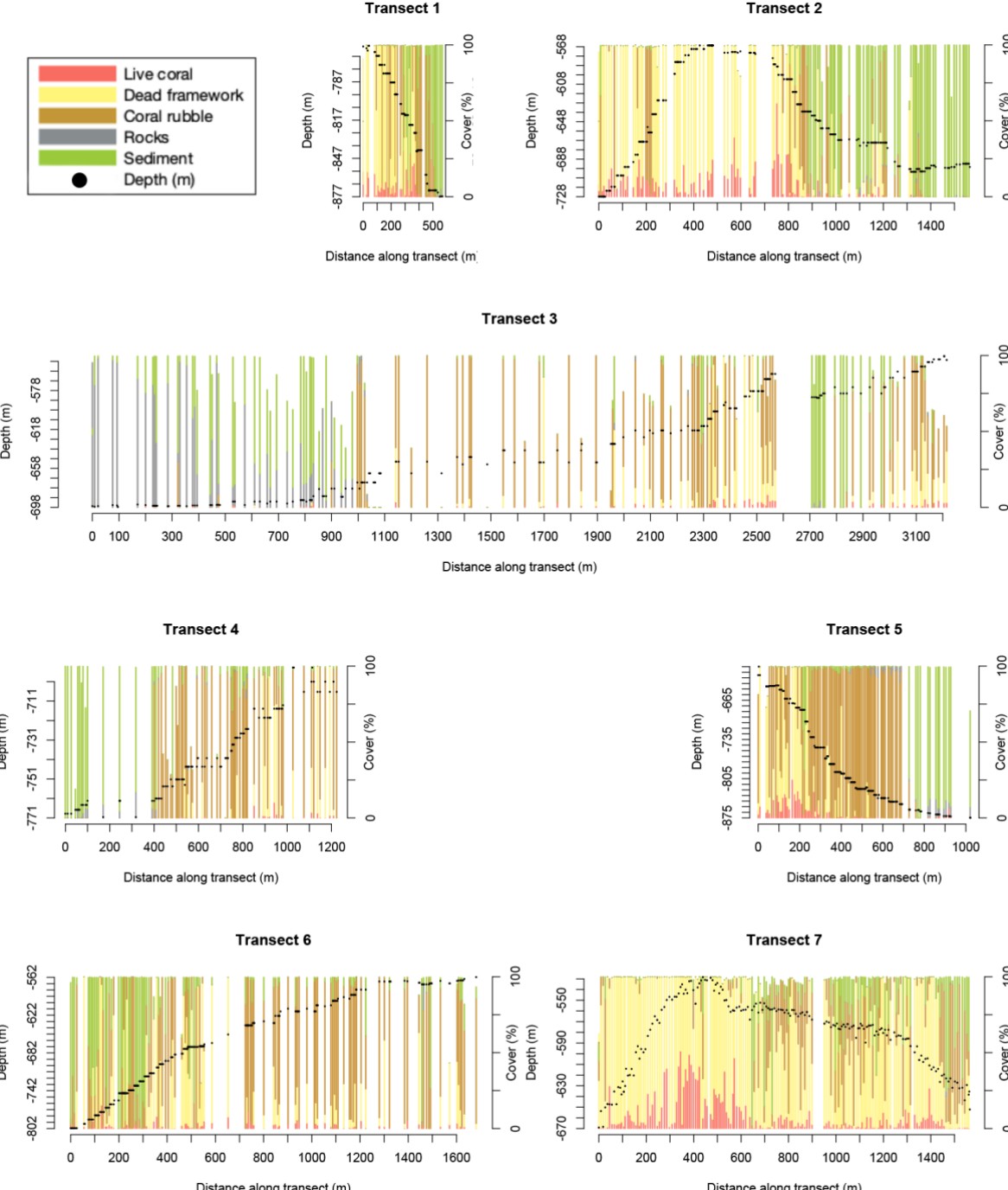

195

**Figure 2. Stacked barplots show the benthic cover (in percent of the annotated frame) of the seven transects. Each bar represents one annotated video frame. Note: for visibility, only frames at >5.5 m distance apart are shown. Benthic cover types shown are the**

cover of live framework-forming corals (red), dead coral framework (yellow), coral rubble (brown), rocks (grey), and sediment (green). Other cover types are not shown, so bar plots might not reach 100 %. Black dots show the transect depth in m (left y-axis).

## 3.2 Redundancy analysis (RDA)

We investigated the relation between benthic cover and environmental variables with a redundancy analysis (RDA) in which we studied the spatial benthic configuration by including Moran's Eigenvector Maps (MEMs) in the analysis. RDA axes 1 to 4 were significant (p<0.001). The first axis of the redundancy analysis explains 27 % of the variation in benthic cover and separates frames with high reef cover (i.e., live corals and dead framework) from frames with low reef cover (i.e., rocks, sediment, and coral rubble cover; Fig. 3 and 4a). The most important variables explaining high versus low reef cover are 'downward velocities in February', the spatial variable 'between-transect MEM 2' and 'depth'. Lesser important variables are 'bottom current speed', 'upward velocities in February', 'between-transect MEM 1', and 'within-transect MEM 4'. The other explanatory variables have only a minor contribution on the first axis. In all, high reef cover is associated with strong downward velocities, shallower water depths, strong bottom currents, and weak upward velocities.

The second RDA axis explains 11.5 % of the variation and separates frames with high coral rubble cover from frames with low coral rubble cover (Fig. 3 and 4b). The most important variable explaining high versus low coral rubble cover is the spatial variable 'within-transect MEM 3'. Lesser important variables are 'between-transect MEM 1', 'upward velocities in August', 'between-transect MEM 2', 'depth', and 'bottom current speed'. High coral rubble cover is associated with strong upward velocities in August, greater depths, and strong bottom current speed.

The third axis explained 6.3 % of the variation in benthic cover (Fig. 4c). The third axis separates frames with high sediment cover from frames with low sediment cover. The most important variables explaining high versus low sediment cover are 'between-transect MEM 1' and 'between-transect MEM 2'. Lesser important variables are 'upward velocities in February', 'downward velocities in August', and 'within-transect MEM 5'. High sediment cover is associated with weak upward velocities in February and strong downward velocities in August. We do not discuss the fourth axis as it explained <1 % of the variation in benthic cover.

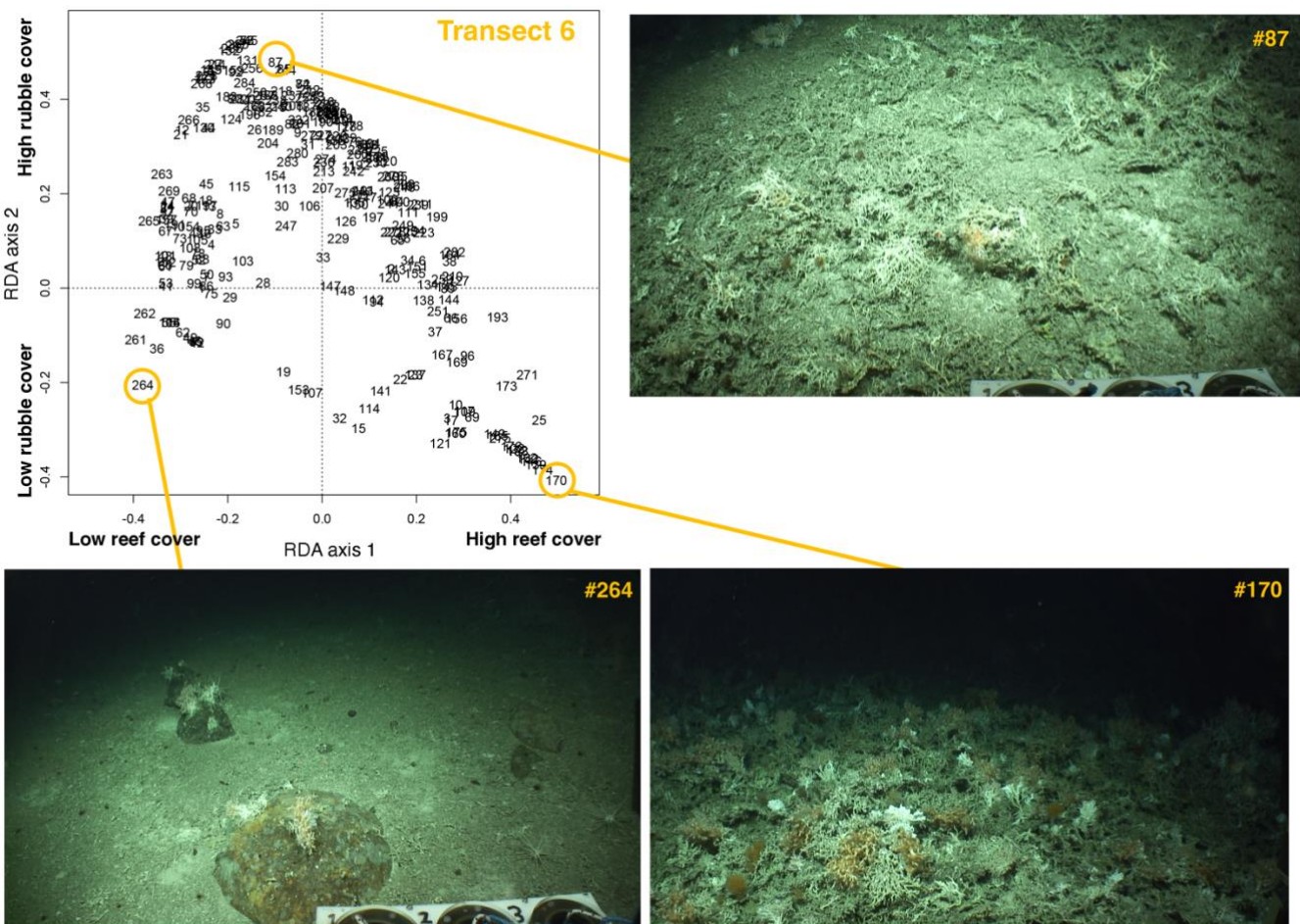

**Figure 3. Annotated video frames from transect 6 plotted on the first (x-axis) and second (y-axis) redundancy analysis (RDA) axes, with three representative images from the seafloor. The first RDA axis separates images with low reef cover (such as image #264) from images with high reef cover (such as image #170). The second RDA axis separates images with low rubble cover (such as image #264 and #170) from images with high coral rubble cover (such as image #87). Note: images are directly extracted from the videos without modifications or annotations. The videos were obtained during the Changing Oceans 2012 expedition with RRS *James Cook* ©2012 Laurence De Clippele.**

225

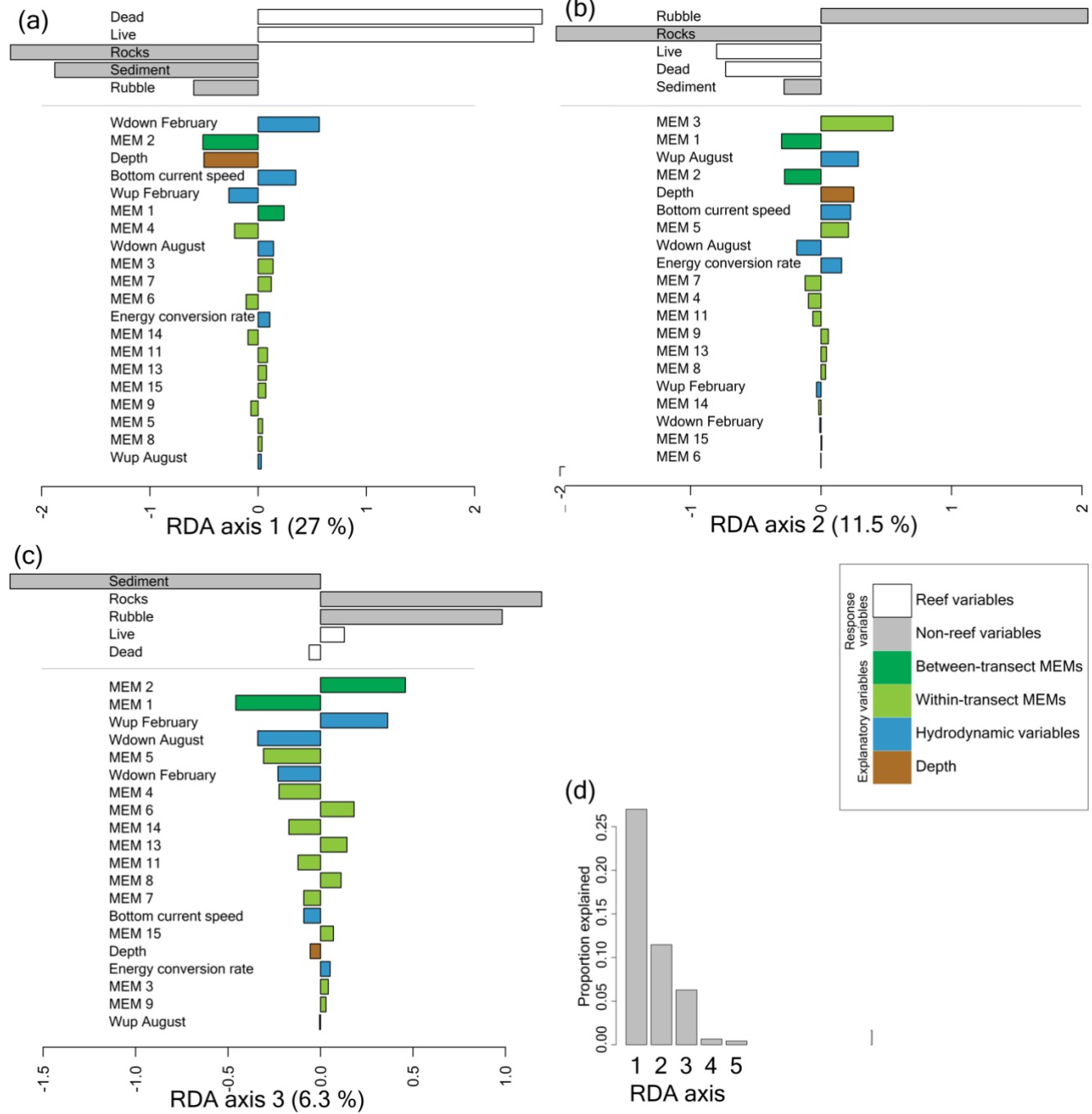

**Figure 4.** Bar plot of the components of the first (a), second (b), and third (c) axis of the redundancy analysis (RDA), explaining 27 %, 11.5 %, and 6.3 % of the variation in benthic cover respectively. The x-axis shows the value on the RDA axis. RDA axes 1 to 4 were significant ($p<0.05$), but axis 4 is not shown as it explained <1 % of the variation in benthic cover (d). The benthic cover (response) variables are shown at the top (reef variables in white and non-reef variables in grey) and below that, the explanatory

**variables are shown in colour. Variables are ordered from most important to least important and coloured according to the variable type as shown in the legend. Response and explanatory variables on the same side of the axis are correlated.**

## 3.3 Variance partitioning

Variance partitioning showed that 39.6 % of the variance in the benthic cover was explained by either depth (11.3 %), hydrodynamic variables (14.2 %), or spatial variables (14.1 %) and 5.5 % by a combination of those variables (Fig. 5). These fractions represent the variation in benthic cover that is explained purely by a (set of) variable(s), meaning that 14.1 % of the variation in benthic cover has a spatial configuration that is not explained by depth or hydrodynamic variables. Of the spatial variables, 2.4 % of the variation in benthic cover is explained purely by between-transect MEMs 1 and 2, i.e., related to variations at the scale of the study area, and 8.8 % by all within-transect MEMs 3 to 15, i.e., related to variations on coral mounds. Since 2.4 % and 8.8% do not add up to 14.1 %, a fraction of the variation in benthic cover (2.9 %) is explained by a combination of between- and within-transect MEMs. A total of 54.9 % of the variance in benthic cover remained unexplained by the explanatory variables used in this study.

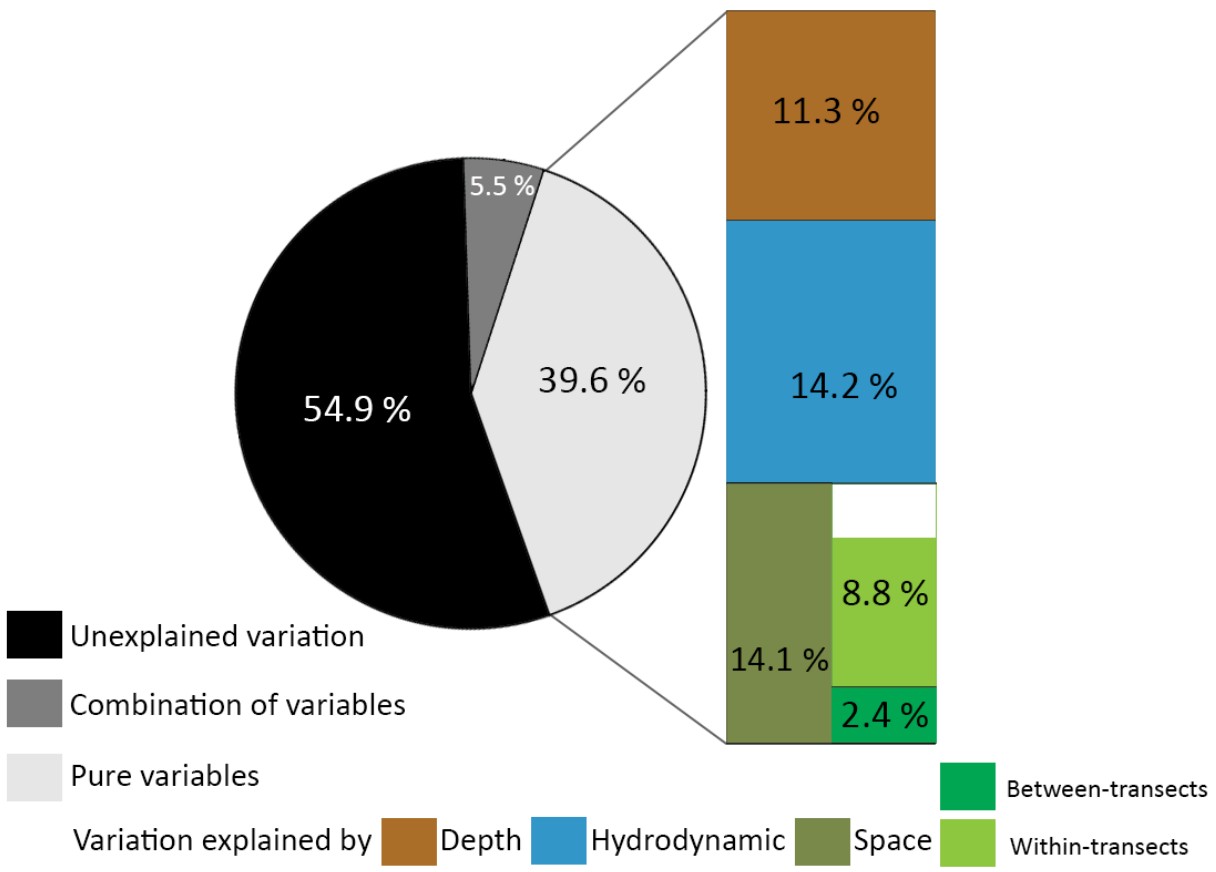

**Figure 5. Pie chart of the proportion of unexplained variation in the data (black), the proportion explained by a combination of sets of variables (dark-grey) and the proportion explained by the explanatory variables (light-grey). The explained variance is separated in the variation explained purely by depth (brown), hydrodynamic variables (blue) or spatial variables (green). Spatial variables are**

further separated in MEMs describing the between-transect spatial scale (dark-green) and the within-transect spatial scale (light-green). The sum of the variance explained purely by the between- and within-transect spatial scale did not add up to 14.1 %, possibly because the combined spatial scales also explain a fraction of the variance. All pure fractions were significant (p<0.001).

### 3.4 Coral mound engineering effect

The comparison of hydrodynamic model output with and without coral mounds from van der Kaaden et al. (2021) showed that the presence of cold-water coral mounds causes an acceleration or deceleration of bottom currents near coral mound foots and
a deceleration on some mound summits (Fig. 6a). Energy conversation rate to the internal tide is increased at those mound flanks that face the tidal current and decreased at the right side of mounds that face the incoming residual current (Fig. 6b). Vertical velocities in August and February are increased in the presence of cold-water coral mounds, downward velocities are increased on mound summits and upward velocities on mound flanks (Fig. 6c-f). Vertical velocities are increased more in February than in August. In August, upward velocities are stronger at the northern mound sides whereas downward velocities
are stronger at the southern side (Fig. 6c-d). There was a significant correlation between mound height and the absolute effect on all hydrodynamic variables, except for downward velocities in February (Fig. 7).

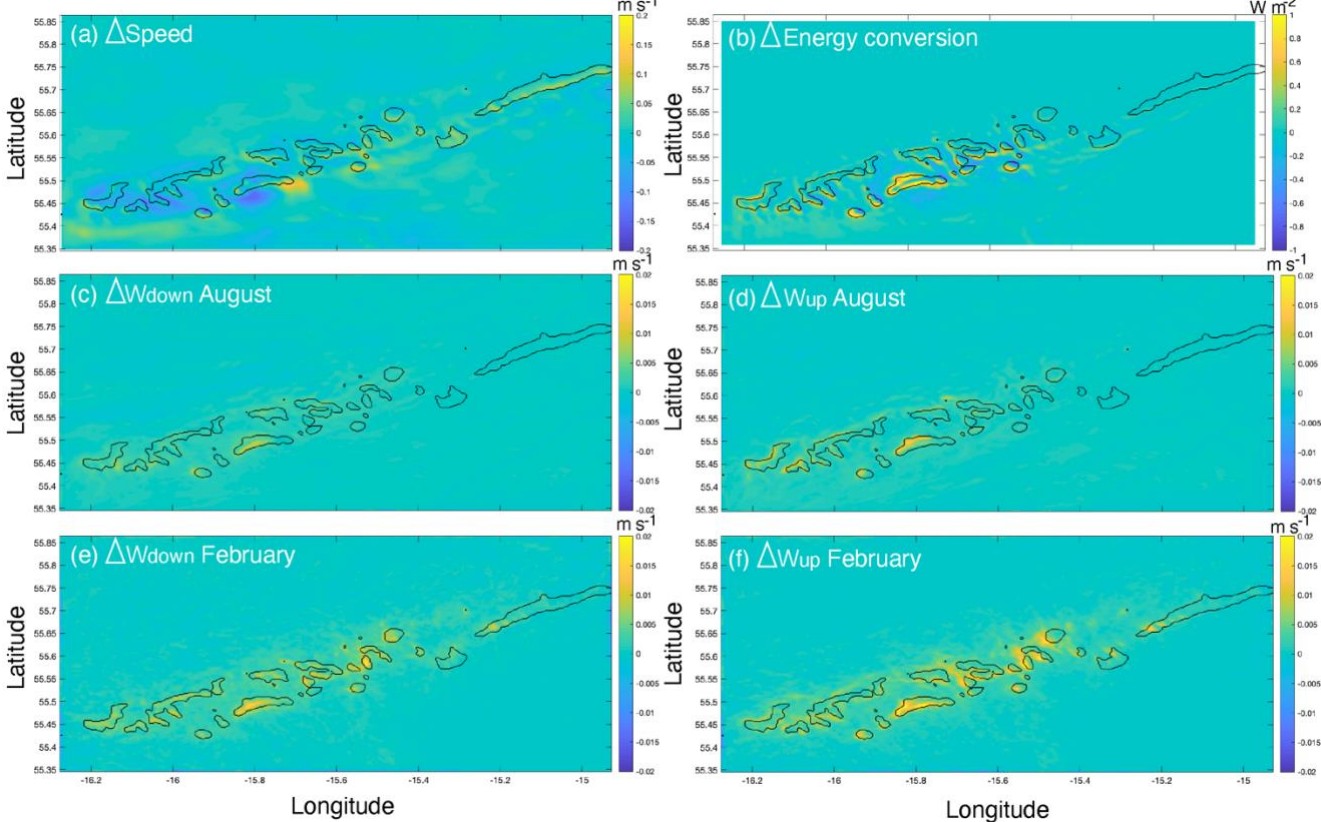

**Figure 6. The effect of the presence of cold-water coral mounds on bottom current speed (a), energy conversion to the internal tide (b), downward velocities in August (c), upward velocities in August (d), downward velocities in February (e), and upward velocities**
**in February (f). The mound effect was calculated by subtracting the hydrodynamic variables (Table 2) obtained from hydrodynamic**

simulations without coral mounds from the variables obtained from hydrodynamic simulations with coral mounds. Positive Δ-values in the panels thus indicate an enhancing effect of coral mounds on the hydrodynamic variables.

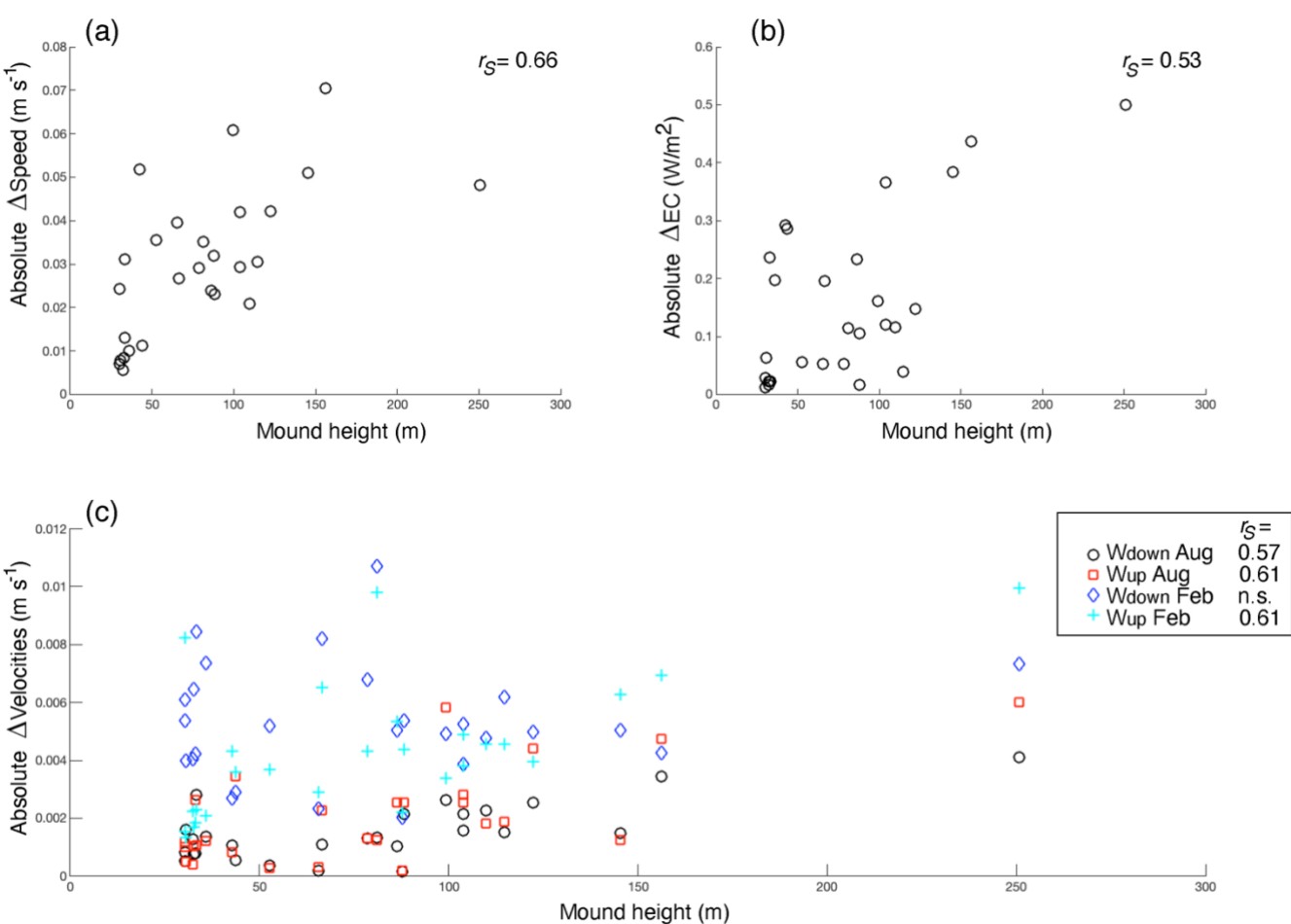

Figure 7. The relationships between mound height and the mean absolute effect of mound presence on (a) the bottom current speed, (b) the energy conversion rate to the internal tide, (c) vertical velocities (i.e., downward and upward velocities in August and February, indicated with different colours and symbols as in the legend). Significant ($p<0.05$) Spearman rho correlation coefficients ($r_S$) for the relationships are indicated in the top right of the figures ($n=27$ mounds), with n.s. denoting correlations that were not significant. Note that the y-axes indicate the effect of mound presence on the hydrodynamic variables, not the actual values.

## 4. Discussion

Through mound formation (ecosystem engineering), cold-water corals can feed higher in the water column. This coral mound engineering also affects the hydrodynamics around coral mounds, possibly increasing the food supply towards the reefs (van der Kaaden et al., 2021; Soetaert et al., 2016). Cold-water corals and most of the fauna associated to the deep-reef habitats rely on organic matter originating from close to the surface ocean (de Froe et al., 2022; van Oevelen et al., 2018; Carlier et al., 2009). Our results underline that, at the Logachev cold-water coral mound province, conditions for cold-water coral reefs are

better higher in the water column since a high reef cover was positively associated to shallower areas (Fig. 2 and 4) and depth explained a large part of the variation in benthic cover (Fig. 5). To determine how coral mound engineering affects the configuration of cold-water corals reefs, we investigated 1) the environmental control on cold-water coral reefs at multiple spatial scales, and 2) which of the environmental factors are/are not engineered (i.e., affected) by the corals (coral mound engineering).

Ventilation and hydrodynamics that stimulate the vertical and lateral food supply have been identified as the most important drivers of cold-water coral reef growth over the past 20,000 years (da Costa Portilho-Ramos et al., 2022). In our analysis we included simulated hydrodynamic variables that indicate increased vertical or lateral food supply, or ventilation. By including spatial variables (Moran Eigenvector Maps, MEMs) in our analysis, we were able to investigate whether these hydrodynamic variables had their largest influence on reef cover at the between- or at the within-transect scale. We found that reef cover

varied most at the scale of the study area (~25 km) and this variation seems associated most to non-engineered (i.e., not under coral control) downward velocities in winter (section 4.1).

By explaining 45.1% of the variation in benthic cover, our set of variables performed well in comparison to similar deep-sea studies using RDA: e.g., Vad et al. (2020) explained 15.1% of the variation in benthic cover and Kazanidis et al. (2021) explained 26.7% of macrofauna variation. Henry et al. (2013) explained 65% of the variation in reef fauna using a combination

of MEMs, hydrography, and bathymetric variables. So, the inclusion of both spatial factors (MEMs) and hydrodynamic variables appears to benefit the analysis. Especially within-transect MEMs described a relatively large part of the variation in our benthic cover data that was not explained by the environmental variables. Henry et al. (2013) similarly found that community composition was structured mainly by broad-scale environmental forcings, but that fine-scale MEMs described a significant fraction of the variation in community composition. This within-transect variation in benthic cover can be related

to spatial patterns of fauna (Henry et al., 2013), such as self-organized regular spatial patterns in coral reefs (van der Kaaden et al., 2023) and to the typical reef zonation on cold-water coral mounds (section 4.2).

The variation that remained unexplained in our study might be related to hydrodynamics at a finer resolution, or other food supply mechanisms that were not described fully by our set of hydrodynamic variables, such as enhanced surface productivity (Eisele et al., 2011; Wienberg et al., 2022, 2020), downward migration of zooplankton (Guihen et al., 2018; Yahel et al., 2005),

other interspecific interactions causing for example recycling of material within cold-water coral reefs (Maier et al., 2020a; Henry et al., 2013; Rix et al., 2018), resuspension (de Froe et al., 2022; Mienis et al., 2009), and particle trapping by the reef framework (Mienis et al., 2019; Maier et al., 2021; Bartzke et al., 2021), or to sediment supply, which is important for reef stabilization and coral mound formation (Bartzke et al., 2021; Pirlet et al., 2011; Wang et al., 2021). Furthermore, all transects included in this study were located on cold-water coral mounds and not in between mounds. Since there are no reefs in between

cold-water coral mounds (Rengstorf et al., 2014), extending transects to include the off-mound region would likely increase the variation explained by the within-transect spatial scale and this could emphasize the coral mound engineering effect.

## 4.1 Broad-scale environmental control on reef cover that is not engineered by cold-water corals

Reef cover (live coral and dead framework) varied most from north to south in the study area. Strong downward velocities in February favour a high reef cover (Fig. 4) likely through an accelerated downward transport of organic matter from near the ocean surface (Soetaert et al., 2016; Frederiksen et al., 1992; Davies et al., 2009; Findlay et al., 2013). Downward velocities in February are stronger with coral mounds present than without coral mounds (Fig. 6c). Modelling the hydrodynamics around one mound of different heights, van der Kaaden et al. (2021) also showed that vertical velocities increased as mound height increased, suggesting a positive feedback mechanism. However, when including all mounds at the Logachev cold-water coral mound province, the magnitude of the effect on downwards velocities in February is not correlated to the height of a coral mound (Fig. 7). We thus argue that high reef cover is associated to enhanced downward velocities mostly due to broad-scale, non-engineered, environmental processes.

Since downward velocities in February did not significantly correlate to downward velocities in August ($r_S$=0.26, $p$=0.18) and downward velocities in August were barely associated to reef cover (Fig. 4), reef cover seems mostly steered by processes prevailing in winter and not in summer. At the Rockall trough, deep winter mixing extends as far down into the water column as 600 m (Holliday et al., 2000) and during winter months dense water is created over the Rockall Bank that cascades down over the Rockall trough margin (White et al., 2005). Both processes cause an accelerated downward flow of organic matter, that peaks near the winter-spring transition, at the onset of the phytoplankton bloom that is particularly early (March/April) at the Rockall Bank (Mohn and White, 2007; Duineveld et al., 2007). These broad-scale, non-engineered, environmental processes offer a food supply mechanism for the cold-water corals during and short after the food-limited winter months (Maier et al., 2020b) and have a larger control on coral survival than food supply mechanisms in summer (Fig. 8a).

This finding suggests that broad-scale environmental processes exert a stronger control on cold-water coral cover than engineered environmental variables. However, the rather coarse resolution of the hydrodynamic simulations might cause us to overestimate the importance of broad-scale environmental variables. Variance partitioning showed that 8.8% of the variation in benthic cover was explained solely by within-transect MEMs, compared to only 2.4% by between-transect MEMs. This signals that the variation at the within-transect scale might be related to environmental variables at a finer resolution. Such fine-scale variation in hydrodynamic variables is more likely to be caused by the interaction between water flow and fine-scale topography, i.e., ecosystem engineering. Since MEMs 3 to 5, that describe the mound foot-to-summit spatial scale, play a role in explaining the variation in benthic cover, cold-water coral mound ecosystem engineering might cause self-organized variation in benthic cover.

Environmental factors other than the hydrodynamic variables investigated here, might also affect coral reef growth and subsequent cold-water coral mound formation. For example, certain water masses (Schulz et al., 2020), the permanent pycnocline (White and Dorschel, 2010), internal waves (Wang et al., 2019; Wienberg et al., 2020), seawater density (Flögel et al., 2014), and (terrestrial) sediment supply (Pirlet et al., 2011; Vandorpe et al., 2017; Lo Iacono et al., 2014) have been suggested to restrain mound formation. Cold-water coral mounds can become buried following changes to the sedimentary

regime (Lo Iacono et al., 2014) and for the Logachev cold-water coral mound province it has been hypothesized that the mounds stopped growing when reaching the permanent pycnocline (White and Dorschel, 2010) or the WTOW upper boundary (Schulz et al., 2020). Van der Kaaden et al. (2021) found no levelling off of the engineering effect of the coral mound on the local hydrodynamics, even when the mound was higher than at present. Such a levelling off is also not apparent in our results (Fig. 6). This underlines that, even though coral mounds engineer their local hydrodynamic environment, coral mound formation could be restricted by non-engineered environmental processes. Still, for cold-water coral mounds that are not buried it can be interesting to investigate the general hydrodynamic regime that arises around cold-water coral mounds and how this regime might explain reef zonation on mounds.

**4.2 Coral mound ecosystem engineering causes self-organized reef zonation**

Here, we demonstrated the effects of coral mound ecosystem engineering: bottom current speeds are accelerated at some parts of the mound and decelerated in other parts. At the mound flank, upward velocities are enhanced, whereas at the mound summits downward velocities are enhanced (Fig. 6). Previous studies (e.g., van der Kaaden et al., 2021; Davies et al., 2009; Findlay et al., 2013; Cyr et al., 2016; Mienis et al., 2007) showed that cold-water coral mounds affect downward water motions. Our results also indicate that coral mounds influence downward velocities in February (Fig. 6). However, the magnitude of this effect does not correlate to mound height, likely because of the prevailing influence of non-engineered broad-scale environmental processes in winter (discussed in section 4.1).

These findings are in line with hydrodynamic theory of waterflow passed an object (Juva et al., 2020; Dewey et al., 2005; Baines, 1995; Genin et al., 1986) and with other studies reporting the hydrodynamics around coral mounds (e.g., Genin et al., 1986; de Froe et al., 2022; Findlay et al., 2013; Davies et al., 2009; Juva et al., 2020). Such a regime can be generalized as: 1) flow acceleration at the mound side facing the incoming current and at the lee, and 2) flow deceleration at the other two mound sides, 3) flow deceleration at the summit if the summit is flat, 4) upwelling along the lower mound flanks, and 5) downwelling around the mound upper flanks and summit. As these typical hydrodynamic zones likely determine the typical benthic cover zones, we will argue in the following that reef zonation on cold-water coral mounds is self-organized.

Cold-water coral mounds show a typical, often reported benthic zonation (Fig. 2), with mostly sediment cover at the mound foot, coral rubble at the lower flank, dead coral framework on the higher flank and living corals at the summit (e.g., Cathalot et al., 2015; Davies et al., 2009; Genin et al., 1986; De Clippele et al., 2021; Maier et al., 2021; Vertino et al., 2014; Freiwald and Roberts, 2005; Dorschel et al., 2005). Van der Kaaden et al. (2023) found self-organized regular patterns in reef cover on the Logachev cold-water coral mounds, with overall reef cover varying along the transects. This suggests that the overall reef cover along a coral mound is steered by environmental conditions that change along the mound, and that, within these benthic zones, cold-water corals self-organize into regular patterns.

Indeed, we found that high cold-water coral reef and rubble cover correlated to the spatial scale from mound foot-to-summit (MEM 3 and 4) and to strong bottom current speeds (Fig. 4), suggesting that reef growth is more prominent on those mound flanks where (engineered) bottom currents are accelerated (i.e., front and rear) than on the flanks where (engineered) bottom

currents are decelerated (Fig. 8b; Mohn et al., 2014). High coral rubble cover correlated to strong upward velocities in August (Fig. 4b), whereas high reef and sediment cover correlated to weak upward velocities in February. This suggests that strong
upwelling occurs mostly around the mound flanks and not around the mound summit and foot (Fig. 8b).

Flow induced up- and downwelling over coral mounds is often mentioned as an important food supply mechanism to cold-water coral reefs (Soetaert et al., 2016; Davies et al., 2009; Wagner et al., 2011; Findlay et al., 2014). During a tidal cycle, isopycnals are depressed at the mound flank and at the turning of the tide an internal bore is formed that propagates over the mound flank towards the summit (Legg and Klymak, 2008; Mohn et al., 2014). The vertical length-scale of these excursions
follows the length-scale of the mound (van Haren et al., 2014; Cyr et al., 2016). Larger coral mounds generally have a larger effect on hydrodynamics than smaller mounds (Fig. 7; Lim et al. 2018; van der Kaaden et al. 2021). This underlines the mound ecosystem engineering aspect of hydrodynamic- and benthic zonation: as the coral mound grows taller, the formerly higher flank regions on the mound that were suitable for coral growth become unsuitable lower flank regions, thereby maintaining the typical reef zonation, and causing a so-called massenerhebung effect (Fig. 8c).

The massenerhebung effect has first been described for the zonation of flora and fauna on terrestrial mountains (Frahm and Gradstein, 1991; Grubb and Whitmore, 1966). Terrestrial mountains exhibit zones with a typical flora and fauna that are determined by various factors such as temperature, humidity, and solar radiation. These zones vary with relative altitude on the mountain, but not with the absolute height above the ground, because of feedbacks between the size of the mountain and the environment (Grubb, 1971; Grubb and Whitmore, 1966). A perhaps surprising result of this effect is that, for example, the
treeline lies higher on higher mountains than on lower mountains. Genin et al. (1986) report that also cold-water coral presence is related to the position on the mound rather than the actual height above the seafloor or depth in the water column. We here hypothesize that this massenerhebung effect, whereby the distribution of organisms is determined by feedbacks between the mountain and the environment, applies to cold-water coral mounds too. Since depth is also an important variable explaining benthic cover, it is likely that the position of cold-water coral reefs on coral mounds is determined by the feedbacks between
the mound and the hydrodynamics, but that shallower mounds have a denser reef cover than deeper mounds.

## 4.3 Ecosystem engineering effects on cold-water corals and climate change

Cold-water coral mounds seem to exhibit a zone with predominantly upward water motions on the mound flanks and a zone with predominantly downward water motions around the mound summit. The summit and upper flank seem most suitable for cold-water coral growth, likely because downward water motions accelerate the downward transport of organic matter. In
contrast, on the coral mound flanks, coral rubble and dead coral framework is more abundant than live corals. Nutrient-rich deep water is pushed up towards the lower mound flanks (Cyr et al., 2016) and the community associated to the dead coral framework releases substantial amounts of nutrients (Maier et al., 2021). These nutrients might be transported by the predominantly upward water motions towards the ocean surface (Soetaert et al., 2016; Findlay et al., 2014; de Froe et al., 2022). This might benefit the coral reefs, as a global review (Maier et al., 2023) and studies on millennial time scales (Eisele
et al., 2011; Wienberg et al., 2022) show the benefits of increased primary productivity for coral reef growth. The upward

water motions around the mound flanks likely also resuspend deposited particles. These particles are transported higher up the mound by lateral advection where they supply the corals with additional food (Findlay et al., 2013; Davies et al., 2009; Mienis et al., 2009).

Other factors than food supply also influence cold-water coral growth, such as biogeochemical seawater properties (Flögel et al., 2014; da Costa Portilho-Ramos et al., 2022; Fink et al., 2012). Deep water that is upwelled around cold-water coral mounds typically has lower temperatures, oxygen content, and aragonite saturation than water from higher in the water column that is downwelled (Findlay et al., 2013, 2014; Mienis et al., 2007; Flögel et al., 2014). The mound zone influenced by downward water motions therefore likely experiences higher temperatures, oxygen, and aragonite saturation state than the region of upward water motions. While higher oxygen and aragonite saturation state would benefit cold-water corals (Fink et al., 2012; Dodds et al., 2007; Tittensor et al., 2009), higher temperatures might have an adverse effect (Morato et al., 2020; Chapron et al., 2021). Currently, most cold-water corals occur within their thermal tolerance (Mienis et al., 2007; Chapron et al., 2021). A higher temperature may be beneficial, but only up to the upper limit of the thermal tolerance range (Büscher et al., 2022; Dodds et al., 2007).

Increased temperatures from global change increase the energy demand of cold-water corals (Dodds et al., 2007; Dorey et al., 2020; Chapron et al., 2021). When not compensated by sufficient food supply, framework-forming cold-water coral habitat is expected to move to deeper waters (Morato et al., 2020). Our results show that this poses a problem to cold-water corals and their associated fauna living on mounds, as the corals prefer the 'downwelling zone' where they profit from a higher vertical particle flux, oxygen content, and aragonite saturation state. The temperatures in this zone, however, might become unfavourable in the future. Further down the mound flank is the 'upwelling zone' that seems less suitable for cold-water coral growth. Studies looking into the geological history of cold-water coral reefs have shown the sensitivity of cold-water coral reefs to a decreased food supply (e.g., Frank et al., 2011; Wienberg et al., 2010; da Costa Portilho-Ramos et al., 2022; Fink et al., 2013; Eisele et al., 2011). This suggest that cold-water corals on cold-water coral mounds might not be able to 'move' down the mound to escape the adverse effects of climate change.

A sufficient food supply may compensate adverse environmental effects up to a certain limit (Büscher et al., 2017; Dorey et al., 2020; da Costa Portilho-Ramos et al., 2022; Hebbeln et al., 2020). But we show that the hydrodynamically stimulated food supply in winter is one of the major factors controlling coral reef cover, at least at the Logachev cold-water coral mound province. Such vertical mixing is thought to dampen with climate change, since stratification might increase (Gerkema et al., 2004; Müller et al., 2014; Pereira et al., 2002), possibly decreasing benthic-pelagic coupling (Capotondi et al., 2012; Li et al., 2018; Reid et al., 2009). So, in the worst possible case, climate change might make cold-water coral mounds entirely less suitable for cold-water coral growth.

## (a) Downward velocities in February

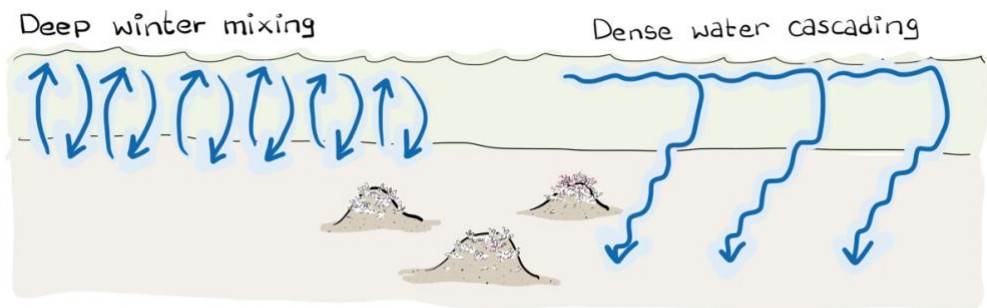

Deep winter mixing                Dense water cascading

## (b) Hydrodynamic zonation

Top view                Side view

accelerated
bottom
currents

reef cover

decelerated
bottom
currents

Coral
mound

downward
velocities

upward
velocities

nutrient
upwelling

mainly ...
live corals

dead corals

coral rubble

Sediment
& rocks

## (c) Massenerhebung effect

**Figure 8. (a) schematic representation of the broad-scale environmental control on reef cover, i.e., downward velocities in February likely resulting from deep winter mixing and dense water cascading. Note that the two processes happen at the same time and place but are here separated in space for clarity. (b) schematic representation of the hydrodynamic zones in a top and side view and the associated benthic zones of "mainly… live corals, dead corals, coral rubble, sediment & rocks". The top view shows that bottom currents are accelerated at the front and rear of the mound and decelerated at the right and left side, looking downstream of the incoming flow. Reef cover is higher on those sides of the mound with accelerated currents and at the mound summit. The side view shows the benthic zones of mainly live corals at the mound summit, dead coral cover at the higher flank, coral rubble at the lower flank and sediment with rocks at the mound foot. The zone of upward velocities and upwelling of nutrients released by the community associated to dead coral framework occurs around the mound flanks whereas downward velocities with transport of organic matter from near the surface happens around the mound summit. The up- and downward water motions do not necessarily happen simultaneously but can result from (spring-neap) tidal motions. (c) schematic representation of the massenerhebung effect on three mounds of different sizes with the benthic zones of mainly live corals (pink), dead coral (light brown), coral rubble (dark brown) and sediment (grey). The massenerhebung effect means that reef cover is determined by the position on the mound rather than the actual height above the seafloor or depth below the sea surface.**

## 5. Conclusions and outlook

We showed that downward velocities in winter, likely driven by non-engineered processes as deep winter mixing and dense water cascading, control reef growth to such an extent that reef cover differs substantially at the scale of the Logachev cold-water coral mound province. These broad-scale hydrodynamically enhanced food delivery processes in winter seem to influence reef cover more than processes related to coral mound ecosystem engineering and are more important for cold-water coral survival than hydrodynamically enhanced food delivery in summer. Nonetheless, by engineering long-lasting mounds, cold-water corals can stimulate the vertical transport of particles from near the ocean surface towards the mound summits. As coral mounds grow, they increasingly affect the hydrodynamics around them and thereby affect cold-water coral reef growth. The initiated hydrodynamic zones shift as the mound grows, leading to a typical reef zonation whereby mound upper flanks and summits have the densest reef cover. Reef zonation on cold-water coral mounds is therefore controlled by coral mound engineering, and reef cover is determined by the position on the mound rather than the actual height above the seafloor or depth in the water column (i.e., the massenerhebung effect).

Much research has shown the benefits of cold-water coral mound engineering for the corals (e.g., Soetaert et al., 2016; van der Kaaden et al., 2021, 2020, 2023; Findlay et al., 2013; Davies et al., 2009), but we here show that the hydrodynamic zones around cold-water coral mounds might restrict cold-water corals to grow deeper on a mound to escape the adverse effects of climate warming. When studying cold-water coral ecosystems and their response to global change the effects of broad-scale, non-engineered environmental processes should not be underestimated, and it is especially important to gain a better understanding of how benthic-pelagic coupling will change in winter and to quantify how much alternative food supply mechanisms and coral mound engineering could compensate a lower food availability.

## Code/Data availability

Data used in this research from (Maier et al., 2021; Raw data for "Reef communities associated with 'dead' cold-water coral framework drive high resource retention and fast recycling in the deap sea") are already published and properly cited in this

paper, and publicly available at https://zenodo.org/record/4076147 . Data used in this research from (De Clippele et al., 2021;

Environmental data and image area measurements of different substrate types extracted from video transects recorded in the

Logachev cold-water coral mound province) are already published and properly cited in this paper, and publicly available at

https://doi.pangaea.de/10.1594/PANGAEA.959612.

**Author contribution**

AvdK, SC, EdF, and DvO were involved in conceptualization and design of the study. LDC, FM, and SRM provided the videos

and annotated images. AvdK, SC, LDC, and SRM were involved in data analysis. DvO supervised the study. DvO, JvdK, MR, and KS were involved in securing funding for this research. AvdK wrote the manuscript. FM collected the video data. All co-authors provided input to the study design, data analysis, data visualization, manuscript writing, and manuscript revisions.

**Competing interests**

The authors declare that they have no conflict of interest.

**Acknowledgements**

This research has been made possible due to collaboration funding between the Royal Dutch Institute for Sea Research and Utrecht University. SRM was funded by the Greenland Research Council. LDC and CM were funded by European Union's Horizon 2020 research and innovation program under grant agreement No. 818123 (iAtlantic). The output of this study reflects only the author's view, and the European Union cannot be held responsible for any use that may be made of the information

contained therein. Funding for video collection was provided by the Netherlands Organisation for Scientific Research NWO-VENI grant 863.11.012.

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
