# Peer review of "Building your own mountain: The effects, limits, and drawbacks of cold-water coral ecosystem engineering"

_EGUsphere, 2023_

## Author Response (AR1)

Thursday 28 September 2023

Author's response to the review of the manuscript titled "Building your own mountain: The effects, limits, and drawbacks of cold-water coral ecosystem engineering."

Dear Peter Landschützer, associate editor with Biogeosciences,

Thank you for the opportunity to resubmit our manuscript after major revisions.

As I understand it, the major concern with our approach is with the percentage of variance explained by the redundancy analysis (RDA). The RDA procedure is a statistical analysis, similar to a principal component analysis (PCA), that is sometimes used in ecological research, but that can be used in any type of data analysis. I would like to highlight that the RDA is a recognized statistical analysis and not something that we invented.

We followed the procedure for the RDA as outlined in the textbook "Ecological Analyses in R" by Borcard, Gillet, and Legendre (2011). They explain that any variable added to the analysis artificially inflates the variance explained. Therefore, from the $R^2$, an adjusted $R^2$ should be calculated that is adjusted for the number of variables used in the test. A forward selection procedure should then be followed that uses the adjusted $R^2$ to test whether an added variable has a real effect on the explained variance or mostly artificially inflates the explained variance. The forward selection procedure stops adding variables to the analysis when an added variable inflates the variance artificially rather than having an important correlation to one of the response variables.

With their example data, Borcard, Gillet, and Legendre explain about 56% of the variance of the response variables in the RDA. They write: "We can be confident that the major trends have been modelled in this analysis. Because ecological data are generally quite noisy, one should never expect to obtain a very high value of adjusted $R^2$."

Another reason that the authors mention to reduce the number of explanatory variables in the RDA is that strong correlations between explanatory variables renders the regression coefficients of the explanatory variables in the model unstable. In other words, the correlations that the RDA would find become unreliable when there are strong correlations between explanatory variables. It is therefore a standard approach to investigate the covariance between explanatory variables prior to the analysis by calculating variance inflation factors (VIFs) and removing those variables that have VIFs above 10. Although, some authors only remove variables with VIFs above 20. We also followed this approach and removed all explanatory variables with VIFs above 11.

We included more explanatory variables in the forward selection procedure that we had available (e.g., turbulent dissipation rates, kinetic energy dissipation rate, and simulated particulate organic matter concentrations). Information about influences such as anthropogenic pressures (e.g., fishing effort) we had not available, and this was also not the focus of our study. Other factors, such as interspecific interactions, random colonization and extinction events, or past disturbances, are simply (nearly) impossible to obtain.

We did not add this information in such detail to our manuscript, as we thought this 'textbook knowledge' would not be very interesting for a typical reader of our manuscript. However, if you think it would be an important addition to our manuscript we can add it to the methods section.

Below, we copied the reviewer comments (in grey) and our detailed response (in black).

In our reply to the reviewers, I wrote that we would try to make a new figure for the appendix to illustrate the MEMs clearer. We tried a few options but found that such figures generally create more confusion than add clarity. We therefore decided to remove the MEMs-figure from the appendix.

Furthermore, kindly, we would like to ask whether we can make a change to the co-author list. As dr. Furu Mienis helped with the manuscript revisions, we would like to add her as a co-author.

Thank you for considering our request and for considering our manuscript for publication in your journal.

With kind regards,

On behalf of all co-authors,

Anna van der Kaaden

Reply to Reviewer 1

The approach of this paper fits in line with several other publications/manuscripts of this group in recent years. It applies statistics to morphological and environmental parameters trying to explain the distribution of cold-water corals on coral mounds or coral mounds in general, in this case referring to the Logachev Mounds at the flank of the Rockall Bank. Actually, most of the data presented here have been published before by these authors, the new aspect here is a (another) different statistical approach to evaluate the data. And I have to admit right away that I am not a specialist in statistics and that it is very difficult for me (if possible at all) to follow the statistical approach applied here (which presumably is due to my ignorance on this topic). That means, I cannot assess the credibility of the statistical work presented here. Nevertheless, probably I am a common "consumer" for such a study with a reasonable background in cold-water coral research that enables me to assess the interpretations and the conclusions presented here.

For many years we already see the self-enhancing effect of coral mound formation. Thus, I applaud the authors for picking this up and trying to go into the details how this process works, which so far was simply seen as a direct link between increasing mound size results in stronger hydrodynamics that result in a better food supply to the corals – with these processes affecting in particular the mound summits. This presence of this general pattern is also described in the conclusions of this ms (although Fig. 2 shows that it does not fully apply to all the mounds investigated here). What is new, is the finding presented here that reef growth is controlled by downward water velocities in 250-300 m water depth in February. Actually, that these downward (and upward) fluxes occur around the mounds has already been presented before (e.g., Sotaert et al. 2016). However, that these control reef growth seems to be a clear overinterpretation of the data. Although this parameter is the strongest contributor to RDA axis 1 (see Fig. 4), it still is only one among several others with a similar weight. Furthermore, RDA axis 1 only explains 27% of the variance, with a total of ~55% being not explained at all by the parameters considered. So, there are probably other parameters involved and the available data do not justify to give the overall control to the downward velocities in February. The mechanistic explanation behind this "control" is seen in "deep winter mixing and deep-water cascading triggering the downward flux of POM (no reference for the POM link!) which is strongest near the winter-spring transition (here: data for February), which fits to the early onset of the phytoplankton bloom (March/April) at Rockall Bank". Mohn and White (2007), referenced for the latter statement, however, show that the spring bloom over Rockall Bank begins in May. Thus, does this mean that the downward fluxes in February predominantly transport "old" POM to the corals or no POM, thereby exerting "strong control" on the coral reef cover. How would this help the corals during the food-limited winter months as stated in line 328? And what happens later in spring, under spring bloom conditions, that most likely provide high amounts of fresh POM. How do the corals react on this? Here it becomes obvious that other parameters (some are mentioned on page 16) need to be included in this study to get a better grip on the controls of benthic cover distribution (and to explain more of the remaining 55% of the variance). Seeing these points, a hydrodynamically stimulated food supply in winter (whereas the model only shows vertical velocities and gives no hint on food supply) as a major factor controlling reef cover (see line 414) is not evidenced by these data.

Dear reviewer 1, thank you for your review of our manuscript. From your comments, we infer that, unfortunately, our main conclusions did not come across clearly enough. We never intended to state that downward velocities in winter resulting from broad-scale non-engineered processes are the only factor controlling reef growth. We added to the revised abstract (line 24): "We find that downward velocities in winter, related to non-engineered environmental factors, e.g., deep winter mixing and dense water cascading, cause substantial differences in reef cover at the broadest spatial scale (20-30 km). Such hydrodynamic processes that stimulate the food supply towards the corals in winter seem more important for the reefs than cold-water coral mound engineering or than similar hydrodynamic processes in summer. While the ecosystem engineering effect of cold-water corals is frequently discussed, our results highlight also the importance of non-engineered environmental processes.

Therefore, we conclude that (line 443): "We showed that downward velocities in winter, likely driven by non-engineered processes as deep winter mixing and dense water cascading, control reef growth to such an extent that reef cover differs substantially at the scale of the Logachev cold-water coral mound province."

And that (line 457): "When studying cold-water coral ecosystems and their response to global change the effects of broad-scale, non-engineered environmental processes should not be underestimated, and it is especially important to gain a better understanding of how benthic-pelagic coupling will change in winter and to quantify how much alternative food supply mechanisms and coral mound engineering could compensate a lower food availability."

That hydrodynamics are important for cold-water corals has been shown, but we extend this knowledge by showing whether it is the engineered hydrodynamics or non-engineered hydrodynamics that exert the major influence on cold-water corals. Moreover, we showed the statistical correlation between cold-water coral cover and hydrodynamic variables whereas earlier studies often remained qualitative (e.g., Soetaert et al. 2016). Our analysis with live and dead coral framework further integrates long-term processes, whereas Soetaert et al. (2016) studied processes on the relatively short term (i.e., tidal processes and spring-neap tide).

We added a paragraph (4.3) on the mechanisms by which the hydrodynamic zones might affect cold-water coral growth.

Indeed, we explain 'only' 45.1% of the variance in benthic cover, which actually means that our RDA performed well in comparison to similar deep-sea studies. The results of an RDA are only reliable when the explanatory variables are not too much correlated to one another. Therefore, we used the forward selection procedure to identify the most important explanatory variables contributing to explaining the variation in benthic cover. Adding more variables would increase the percentage of variance explained but would make the analysis unreliable.

Other variables that might influence benthic cover are anthropogenic influences, small-scale differences in environmental variables, interspecific interactions, or random effects, but these could unfortunately not be included in this study.

To the discussion (line 294) we added: "By explaining 45.1% of the variation in benthic cover, our set of variables performed well in comparison to similar deep-sea studies using RDA: e.g., Vad et al. (2020) explained 15.1% of the variation in benthic cover and Kazanidis et al. (2021) explained 26.7% of macrofauna variation. Henry et al. (2013) explained 65% of the variation in reef fauna using a combination of MEMs, hydrography, and bathymetric variables. So, the inclusion of both spatial factors (MEMs) and hydrodynamic variables appears to benefit the analysis. Especially within-transect MEMs described a relatively large part of the variation in our benthic cover data that was not explained by the environmental variables. Henry et al. (2013) similarly found that community composition was structured mainly by broad-scale environmental forcings, but that fine-scale MEMs described a significant fraction of the variation in community composition. This within-transect variation in benthic cover can be related to spatial patterns of fauna (Henry et al., 2013), such as self-organized regular spatial patterns in coral reefs (van der Kaaden et al., in press) and to the typical reef zonation on cold-water coral mounds (section 4.2). "

Mohn and White (2007) describe Chl-a levels over the Rockall Bank that are enhanced throughout the year and remain high during winter as compared to the surrounding water. In their figure 3, Chl-a levels over the Rockall Bank are higher in March than in February. Duineveld et al. (2007) further report that a fresh phytoplankton signal reaches the cold-water coral mounds at the end of February, signalling the start of the phytoplankton bloom. We added the reference to Duineveld et al. (2007) and changed the sentence to (line 327) to: "Both processes cause an accelerated downward flow of organic matter, that peaks near the winter-spring transition, at the onset of the phytoplankton bloom that is particularly early (March/April) at the Rockall Bank (Mohn and White, 2007; Duineveld et al., 2007)."

Consequently, in its present stage this manuscript does not provide any new, well documented information on the controls on benthic cover in the Logachev Mounds. Thus, this ms definitely needs to consider further parameters, add more process understanding, provide more information about the meaning of its MEMs, and discuss its new ideas (rather than just backing them up by references that do not really fit).

Further points to consider are listed below:

Coral distribution: In line 193 it is stated that the live coral coverage typically increases towards the mound summit. The use of the term "typically" already hints to a somewhat weak signal. And indeed, in Fig. 2 in transect 1, highest live coral coverage is at the lower slope and in transects 2 and 5 it is along the upper slopes. In transects 3, 4, and 6 live coral coverage is continuously very low with maybe some more corals at the summit in transect 3. Thus, a clear signal reflecting the statement above is only seen in transect 7, thus, 1 out of 7. One might include here transects 2 and 5 with highest coral coverage along the upper slopes, but these do not show any further increase towards the summit. So, basically, the statement repeated above is somewhat misleading – as already seen by the authors letting

them put in "typically". In the conclusions, it is again referred to that only mound summits are topped with dense live corals (line 395).

We phrased the sentence as we did because we thought that clear trends were not obvious from the raw data. We now rephrased the sentence to (line 190): "There is variability in benthic cover within- and between transects (Fig. 2), but clear trends are not obvious. For instance, transect 3, located on the lower southern flank of the northernmost mound, and transect 6, located at the western flank of the highest mound (Fig. 1), have a relatively low live and dead coral cover overall. Transect 7, across the highest mound in the area, shows a relatively high dead coral cover and low sediment and rubble cover overall."

We also rephrased the sentence in the conclusion (line 450): "…, leading to a typical reef zonation whereby mound upper flanks and summits have the densest reef cover."

Interestingly, the RDA reveals that the most important single variable to explain high vs. low reef cover between 600 and 1100 m water depth is the "February downward velocity between 250 m and 300 m water depth". In addition, for the 2$^{nd}$ RDA axis, differentiating areas with high vs. low coral rubble cover, the spatial variable within-transect MEM3 dominates. The 4$^{th}$ RDA axis, finally differentiating between live and dead coral cover, however, only explains <1% of the variability and is driven by within-transect MEMs 6 and 7 and by "upward velocities in February". Thus, in addition to the up- and downward velocities, the MEMs are important variables controlling the benthic cover. However, besides one very vague sentence (line 299-301), no hint is given in which way the MEMs exert any control. If these are so important, a clearer definition of the MEMs and an in-depth discussion how (!) these affect the benthic cover would be needed. Fig. A1 is of no help in this context: "colours and bubble size represent the values belonging to the individual annotated video frames" – which values? which of the 1200 video frames? how do colours and bubble sizes related to any numbers? I am sorry, but I cannot read anything out of this figure. This figure is also referred to in line 350ff, where in combination with Fig. 4 evidence is presented that high coral reef cover is driven by MEM 3 and 4 and by strong bottom currents. However, Fig. 4 points to downwelling in February, MEM2 and depth as most prominent parameters. It also does not become clear to what extent MEM3 and 4 are linked to specific flanks of the mounds … This part is quite confusing.

MEMs merely describe a certain spatial scale of spatial separation between the observation points, so they do not exert any control on reef cover. We clarified this by changing section 2.2.3 (line 136): "In the MEM-approach, the study area is divided into groups (i.e., a map with 1 group, 2 groups, 3 groups, etc.) and each annotated video frame is assigned a value that represents its proximity to the centre of the group in which it is located (Borcard et al., 2011). Moran's Eigenvector Map 1 thus represents variations on the broadest spatial scale of the entire study area (~25 km) and the highest MEMs (i.e., the when the study area is divided in the highest number of groups) describe variation at the finest spatial scale of the difference between individual annotated video frames (~4m). We visualized the MEM-procedure in see Fig. A1 (Appendix A). MEMs merely describe a certain spatial scale and thus do not exert any control on CWC growth, their specific values are therefore meaningless with respect to CWC growth."

And (line 146): "At the broadest spatial scale (signalling differences between transects) datapoints were classified according to their east-to-west position (MEM 1) or according to

their north-to-south position (MEM 2). At finer scales (signalling differences within transects) datapoints were classified by MEMs according to their position along a mound flank (MEM 3, 4, and 5), or according to regular intervals along transects (MEM 6 to MEM 15)."

To clarify the discussion section 4.2, we changed the sentence in line 360: "…we found that high cold-water coral reef and rubble cover correlated to the spatial scale from mound foot-to-summit (MEM 3 and 4)…". We further removed the reference to the supplementary figure in the discussion.

We will clarify the supplementary figure illustrating the MEM values: showing the values of MEM 1 to 7, along three example transects for each annotated video frame.

This continues in lines 354 ff. While the first sentence is not to follow as the described settings are not obvious in Fig. 4, it does not become clear at all, why these suggest strong upwelling around the mound flanks … Most of the contribution of the parameters described in the first sentence to the RDA axes are really minor. Thus, to me also this reads as a clear overinterpretation of the data. And the following sentence (line 356) is pure speculation (especially as the last two references only refer to the benefit high primary productivity can have for the corals).

Indeed, the sentence was ambiguous. We changed it to (line 363): "High dead coral cover correlated to strong upward velocities in February (section 3.2) and high coral rubble cover to strong upward velocities in August (Fig. 4b), whereas high live coral and sediment cover correlated to weak upward velocities in February (section 3.2)."

The fourth RDA axis contributes significantly, albeit explaining a smaller fraction of the variance than the other axes. We moved the sentence of the upwelling of nutrients to the new discussion section 4.3 and added references on the nutrient upwelling (line 391): "Nutrient-rich deep water is pushed up towards the lower mound flanks (Cyr et al., 2016) and the community associated to the dead coral framework releases substantial amounts of nutrients (Maier et al., 2021). These nutrients are likely transported by the predominantly upward water motions towards the ocean surface (Soetaert et al., 2016; Findlay et al., 2014; de Froe et al., 2022) where they stimulate primary productivity that, in turn, benefits the coral reefs (Eisele et al., 2011; Wienberg et al., 2022, 2020)."

Mound engineering effect: a key finding here is that the mound engineering effect results in a deceleration on mound summits (line 256), which is picked up again in the discussion (line 332). So, why do the corals prefer the mound summits? Available current meter data usually show highest current speeds at or close to the summit. So, what is the meaning of your assessment that the mound engineering leads to reduced current speeds at the summits? And in Fig. 7a it is shown that bottom current speed significantly increases with mound height. But this does not refer to the summits? From line 333 this continues with a description of a hydrodynamic regime on coral mounds also seen in other studies. However, neither in Genin et al. nor in Soetaert et al., this pattern becomes obvious. See, for instance, points 1) and 2), for which neither of the two other paper present any evidence, and the same applies partly to the other points. These are to see to some extent in the modelling study by van der Kaaden et al. (2021), however, field observations do not show such a pattern (e.g., Genin et al. 1986, Dorschel et al., 2007). So, the pattern described here might be introduced as a new conceptual idea, but then it needs more discussion about the

involved processes, and not simply the hint to some other papers (which actually do not show this pattern).

Horizontal bottom currents are only decelerated when mound summits are flat. We write (line 257): "…a deceleration on *some* mound summits". In the discussion we refer to it again (line 351): "…, 3) flow deceleration at the summit if the summit is flat,…".

Figure 7a does not show that horizontal bottom currents increase with mound height, but that the absolute *effect* on bottom current speed increases with mound height. To clarify we added to the caption of figure 7: "Note that the y-axes indicate the effect of mound presence on the hydrodynamic variables, not the actual values."

The description of the hydrodynamic regime is based on Genin et al. (1986) where it is eloquently explained: "The distribution of corals on wide peaks can be explained by prediction of physical theory. When water is upwelled above wide topography, vortex lines are compressed and anticyclonic motion is induced, because of conservation of potential vorticity. The resulting flow-field is a combination of the velocity induced by this anticyclonic component and the free-stream velocity. Thus, looking downstream, flow acceleration occurs on the left side of the topographic feature (in the Northern Hemisphere) and deceleration above the centre and right side of the peak. […] The overall mean velocity pattern would therefore comprise alternating periods of acceleration on the peak's edges and recurrent deceleration trend over the centre. This postulated velocity pattern is consistent with the lower densities of corals observed at the centre of the wide peaks of Jasper Seamount. Furthermore, the combination of tidal fluctuations and a general southward flow in the vicinity of the seamount could result in a stronger average acceleration on the eastern side of the seamount."

The upwelling along the flanks and downwelling at the summit can be seen in Soetaert et al. (2016) figure 3B where the downwelling is also explicitly linked to the organic matter flux (their figure 3F). We substantiated the references to the hydrodynamics and rephrased (line 345): "These findings are in line with hydrodynamic theory of waterflow passed an object (Juva et al., 2020; Dewey et al., 2005; Baines, 1995; Genin et al., 1986) and with other studies reporting the hydrodynamics around coral mounds (e.g., Genin et al., 1986; de Froe et al., 2022; Findlay et al., 2013; Davies et al., 2009; Juva et al., 2020)."

One comment on the methodology: what does not become clear from the text, is how the hydrodynamic model data with grid cells of 250 m in the horizontal are combined with the video frame data. According to Fig. 2, the individual transects are between ~500 and ~1500 m long; only transect 3 is much longer with ~3100 m, but has a less dense video frame density. Thus, most transects correspond to 2 to 6 grid cells in the hydrodynamic model, with seeing the small-scale typical summit and slope settings probably being partly very difficult to discern within 250 m grid cells. On the contrary, the video frame density usually is much, much higher, with a total of ~1200 frames. Thus, ~1200 frames, which in parts show quite some variability along a transect, are linked to ~30 grid cells, that seemingly cannot always discern between summits, slopes, etc. – how this is done becomes – at least for me – not clear from chapter 2.3. Chapter 2.2.2 explains that for each frame the nearest grid cell

has been chosen for correlation, so, there might be a few more than the 30 grid cells mentioned above. But nevertheless, the question remains …

Indeed, the hydrodynamic simulations have a rather coarse resolution which may cause us to overestimate the importance of the broad-scale non-engineered hydrodynamics in the RDA. We did not stress this enough in our manuscript, so to section 4.1 we added (line 332): "This finding suggests that broad-scale environmental processes exert a stronger control on cold-water coral cover than engineered environmental variables. However, the rather coarse resolution of the hydrodynamic simulations might cause us to overestimate the importance of broad-scale environmental variables. Variance partitioning showed that 8.8% of the variation in benthic cover was explained solely by within-transect MEMs, compared to only 2.4% by between-transect MEMs. This signals that the variation at the within-transect scale might be related to environmental variables at a finer resolution. Such fine-scale variation in hydrodynamic variables is more likely to be caused by the interaction between water flow and fine-scale topography, i.e., ecosystem engineering. Since MEMs 3 to 5, that describe the mound foot-to-summit spatial scale, play a role in explaining the variation in benthic cover, cold-water coral mound ecosystem engineering might cause self-organized variation in benthic cover. "

Obviously, the authors like their own phrases very much. For instance, the hint to better conditions for the corals "higher up in the water column" occurs at least 4 times, e.g., with the statement that the corals would benefit from being higher in the water column as there, POM quality and quantity is higher (only giving a reference for the POM distribution, but not for the link to the corals). This is followed up at the beginning of the discussion where coral performance again is linked to water depth – described as a result of this study. However, these statements are in contrast to one of the main conclusions referring to the "Massenerhebung effect" stating that not the position in the water column, but the position on the mound is a key parameter controlling coral growth. In addition, also the title of the ms is more a nice wording than a good description of the content of the paper. While the word "mountain" in the first part is only picked up in the paper with reference to terrestrial mountains, the second part of the title goes far beyond its actual content.

Depth explained 11.3% of the variation in benthic cover and was an important variable in the RDA axis 1 and 2, so that depth is an important variable explaining benthic cover is a finding of our study. As cold-water coral reef cover can be more or less dense, this result is not in contrast with the massenerhebung effect. To the discussion (4.2) we added (line 384): "Since depth is also an important variable explaining benthic cover, it is likely that the position of cold-water coral reefs on coral mounds is determined by the feedbacks between the mound and the hydrodynamics, but that shallower mounds have a denser reef cover than deeper mounds."

Table 1 caption: which colours?

We removed the reference to colours from the caption.

Line 177: with (!) coral mounds

Yes, thank you.

Review ms from Van der Kaaden et al, Building your own mountain: the effects, limits, and drawbacks of cold-water coral ecosystem engineering

The work of Van der Kaaden et al, in Biogeosciences egusphere-2023-949 provides a study of the Logachev cold-water coral mound province to investigate their ecosystem engineering ability at a local scale. To do so, they use 1) hydrodynamics modelling to test the effect of presence/absence of the coral mounds in currents and energy dissipation, and 2) statistical analysis (redundancy analysis) based on video transect images and environmental variables (both hydrodynamics and spatial variables from Moran's Eigenvector Maps).

Major comments:

Self-structuration of the CWC reefs and their environmental drivers is an important question to assess especially in a global change context where sustainable management and future remediation strategies are to be defined, and the ms from Van der Kaaden et al, has the advantage to try to tackle this issue. The ms is well written and properly referenced. However, I feel like the ms in its current form needs further work before it can be published in Biogeosciences.

One of the difficulty I have with the ms is that some of the conclusions seem speculative, and the authors need to better assess their conclusions, their reliability to be more assertive.

For starter, their RDA is based on variables that fail to explain 55% of the variance, leaving a significant part of the variance in the CWC benthic cover unexplained: among the principal contributors are depth, and hydrodynamic parameters (which had already been demonstrated see Wagner et al, 2011, De Clippele et al, 2018, Mohn et al, 2014, Cyr 2016, Van der Kaaden et al, 2012 – all referenced in the ms), and the first four MEMS.

Dear reviewer 2, thank you for your helpful comments. The results of RDA are only reliable when the explanatory variables are not too much correlated to one another. Therefore, we used the forward selection procedure to identify the most important explanatory variables contributing to explaining the variation in benthic cover. Adding more variables would increase the percentage of variance explained but would make the analysis unreliable.

Compared to similar deep-sea studies using an RDA on benthic cover, our analysis actually performed better or comparable. Other factors that might influence benthic cover are anthropogenic influences, small-scale hydrodynamics, interspecific interactions, or random effects, but these were not the focus of our study. That hydrodynamics are important for cold-water corals has been shown before, but we extend this knowledge by showing whether it is the 'engineered' hydrodynamics or 'non-engineered' hydrodynamics that exert the major influence on cold-water corals. In contrast to earlier studies (e.g., Wagner et al. 2011, De Clippele et al. 2017, Cyr et al. 2016, and van der Kaaden et al. 2021), we identified significant statistical correlations between cold-water coral cover and hydrodynamic variables.

To the discussion (line 293) we now added: "By explaining 45.1% of the variation in benthic cover, our set of variables performed well in comparison to similar deep-sea studies using RDA: e.g., Vad et al. (2020) explained 15.1% of the variation in benthic cover and Kazanidis et al. (2021) explained 26.7% of macrofauna variation. Henry et al. (2013) explained 65% of the variation in reef fauna using a combination of MEMs, hydrography, and bathymetric variables. So, the inclusion of both spatial factors (MEMs) and hydrodynamic variables appears to benefit the analysis. Especially within-transect MEMs described a relatively large

part of the variation in our benthic cover data that was not explained by the environmental variables. Henry et al. (2013) similarly found that community composition was structured mainly by broad-scale environmental forcings, but that fine-scale MEMs described a significant fraction of the variation in community composition. This within-transect variation in benthic cover can be related to spatial patterns of fauna (Henry et al., 2013), such as self-organized regular spatial patterns in coral reefs (van der Kaaden et al., in press) and to the typical reef zonation on cold-water coral mounds (section 4.2)."

 I find it hard to identify what the MEMs can realistically represent. Especially since mainly the larger ones are significant contributors to the RDA 2 main axis (MEM 1 to 4) which therefore corresponds to what? Several km scale? If MEM1 is 25km entire study area, .MEM4 is down to which spatial scale? Figure Appendix A1 lacks a color and a bubble scale to help interpret more easily these spatial variables. Especially since the conclusions of the authors are made on the mound scales – mound being in the order of hundreds of metres to a few kilometres. Maybe provide scalograms to indicate the portion of variance (R2) explained by each spatial scale (See Dray et al, 2012 https://doi.org/10.1890/11-1183.1). Regarding this spatial scale consideration, two remarks:

Hydrodynamic simulations: the grid size being 250m, how confident are the authors to mimic and interpret the effect of mound removal when mound sizes are "in the order of hundreds meters to a few kilometers (Wheeler et al, 2006 https://link.springer.com/article/10.1007/s00531-006-0130-6)" mainly "up to 100 m high, 1–2 km long "(Van Weering et al., 2003)? Is 2 to 4 (max 8) grid points enough to draw hard conclusions? Did the authors perform any sensitivity analysis (maybe testing the minimum mound size to be have an effect on the hydrodynamics variables they consider)? Just to give the reader the sense on how the hydrodynamics effect of the CWC presence act.

In Van der Kaaden et al. (2021), we investigated the effect of cold-water coral mounds of different sizes on the hydrodynamics and showed that mounds influence the hydrodynamics already from about 25m height. By correlating the height of all Logachev mounds to their effect on the hydrodynamics we here show that all hydrodynamic variables are significantly affected by cold-water coral mound presence, except for downward velocities in February. For this correlation, the scale of the hydrodynamic simulations is of lesser importance as we calculated the mean values for each mound.

In the RDA, the scale of the hydrodynamic simulations is more important and might lead to an overestimation of the importance of broad-scale environmental processes over finer-scale processes. Indeed, we did not stress this enough in our manuscript (also mentioned by reviewer 1), so to section 4.1 we added (line 332): "This finding suggests that broad-scale environmental processes exert a stronger control on cold-water coral cover than engineered environmental variables. However, the rather coarse resolution of the hydrodynamic simulations might cause us to overestimate the importance of broad-scale environmental variables. Variance partitioning showed that 8.8% of the variation in benthic cover was explained solely by within-transect MEMs, compared to only 2.4% by between-transect MEMs. This signals that the variation at the within-transect scale might be related to environmental variables at a finer resolution. Such fine-scale variation in hydrodynamic variables is more likely to be caused by the interaction between water flow and fine-scale topography, i.e., ecosystem engineering. Since MEMs 3 to 5, that describe the mound foot-to-summit spatial scale, play a role in explaining the variation in benthic cover, cold-water coral mound ecosystem engineering might cause self-organized variation in benthic cover."

If the four first MEMS are the main spatial non-hydrodynamic environmental drivers of the CWC benthic cover, and therefore represent spatial variations "between transect" and "within transect MEM3 and 4" of several hundred meters, how can the authors draw conclusions at the mound scale? Although I am rather seduced by the masserhebung explanation and the analogy to mountain specie zonation, I fail to reconcile the scales at stake there.

MEM 3, MEM 4, and MEM 5 describe the variation at the scale from mound foot to mound summit. MEM 3 is the most important explanatory variable explaining the variation in rubble cover versus non-rubble cover (RDA axis 2) and MEM 4 plays a role in explaining the variation in coral reef versus non-reef cover (RDA axis 1). Between-transect MEMs 1 and 2 describe more of the variation in the RDA, but variation partitioning shows that a larger share of the variance in benthic cover is explained solely by within-transect MEMs (i.e., MEM 3 and 4 and to a lesser extent 5, 6, and 7). This indicates that the mound foot-to-summit variation in benthic cover is, for quite a significant part (i.e., almost as much as depth), not explained by environmental variables included in this study. Spatial variation that cannot be related to environmental variables can be self-induced, as with a massenerhebung effect. It is not unreasonable to think that a massenerhebung effect might happen at cold-water coral mounds, which is why we propose that a massenerhebung effect exists for cold-water coral mounds.

To clarify the scale described by the different MEMs, we changed section 2.2.3 (line 146): "At the broadest spatial scale (signalling differences between transects) datapoints were classified according to their east-to-west position (MEM 1) or according to their north-to-south position (MEM 2). At finer scales (signalling differences within transects) datapoints were classified by MEMs according to their position along a mound flank (MEM 3, 4, and 5), or according to regular intervals along transects (MEM 6 to MEM 15)."

We will clarify the supplementary figure illustrating the MEM values: showing the values of MEM 1 to 7, along three example transects for each annotated video frame.

And in line 360: "…we found that high cold-water coral reef and rubble cover correlated to the spatial scale from mound foot-to-summit…" instead of "…the spatial scale of a mound…".

We further worded our conclusions more carefully (line 382): "We here hypothesize that this massenerhebung effect, …, applies to cold-water coral mounds too." Instead of: "We here present evidence…"

Detailed comments:

Line 179 add positive in "i.e.a feedback". As feedbacks can be both ways…

Yes, that is true, but we intend to refer to a feedback (writing that the effect of the mound on the hydrodynamics increases with increasing mound height). At some sides of the mound bottom currents are decelerated, so we refer to both a positive and negative feedback.

Figure 3: Images #81 and #52 are blurry. Maybe the authors could select other more focused images to make their point alhtough I do understand the need to select "end-members" pictures in terms of reef cover and rubble abundance.

We used higher-resolution images from a different transect for figure 3.

Figure 8b) Side view – it is rather difficult to picture a convection loop with descending currents and ascending currents concomitant at that scale of a mound… Do the authors have

any measurements (or model output) of such co-existing opposite vertical currents?

Indeed, such 'upwelling' and 'downwelling' can be seen in measurements and model output (e.g., Soetaert et al. 2016, Davies et al. 2009, Cyr et al. 2016), and is known from theory of waterflow around an obstacle (e.g., Juva et al. 2020). Of course, the up- and downward motion do not necessarily exist at the same time, but can be the result of tidal processes. We clarified in the caption of figure 8: "The up- and downward water motions do not necessarily happen simultaneously but can result from tidal motions."

Line 320-330. Only transects 2, 3, 6 and 7 reach 600m, transects 1, 4 and 5 are deeper: the deep water mixing extending down to 600m won't reach these mounds. Please suggest additional mechanisms for food supply to those.

Deep-water mixing brings organic matter from the ocean surface some hundred meters down into the water column. This would increase the organic matter quality and quantity at all depths not only within the mixed layer. So, the mechanisms that we propose are relevant also for the deeper mounds.

Line 354 – 355. The authors state "High dead coral cover correlated to strong upward velocities in February and high coral rubble cover to strong upward velocities in August, whereas high live coral and sediment cover correlated to weak upward velocities in February (Fig. 4).". Although I do see correlation for high coral rubble cover to strong upward velocities in August, I failed to see the two others. Both high dead coral and live coral cover seem correlated to strong downward February (Wdown February being on the same side of the axis that Dead and Live coral). Correlation between sediment and weak upward velocities in Feb also seems ambiguous.

Indeed, this was unclear. The reference to figure 4 concerned only the coral rubble distinction from RDA axis 2. We changed the sentence (line 363): "High dead coral cover correlated to strong upward velocities in February (section 3.2) and high coral rubble cover to strong upward velocities in August (Fig. 4b), whereas high live coral and sediment cover correlated to weak upward velocities in February (section 3.2)."

Line 360-363. It is worth mentioning that hydrodynamics (upwelling and downwelling) and tidal cycles impact also the biogeochemical properties of the seawater (e.g. inorganic carbon and pH) above the reef and therefore the ecosystem functioning of the reef (Findlay et al, 2013 https://doi.org/10.1111/gcb.12256 Jiang et al, 2020 https://doi.org/10.1016/j.ocemod.2019.101555 , Findlay et al, 2014 https://doi.org/10.1038/srep03671). Maybe the authors should elaborate on that and integrate it to the discussion.

Yes, we like this suggestion very much, so we added a section (4.3) on "Ecosystem engineering effects on cold-water corals and climate change", where we discuss the mechanisms by which the hydrodynamic zones might affect cold-water coral growth, including temperature, oxygen, and pH.

---

## Referee Report (RR1)

**Review EGUSPHERE manuscript 2023-941**

**Title:** Building your own mountain: The effects, limits, and drawbacks of cold-water coral ecosystem engineering

**Author:** Anna-Selma van der Kaaden et al.

**General remarks:**

The manuscript went already through a review process with suggested major revisions and the present revised version improved a lot. However, I still have some concerns, which I would like the authors to consider:
- I think the authors should better differentiate between short and long-term processes in their discussion, please see detailed comment for lines 388–398;
- The authors could better discuss the relationship of mound height to increasing hydrodynamic variables (see more detailed comment to Lines 181 and 281 below). I miss the discussion on mounds reaching a maximum height at the boundary (perm. thermocline, water mass, density) and their behavior/control at the maximum height, or mounds occurring at much shallower water depths (e.g., Norway);
- Figures and tables could be improved, see comment to Fig. 1, Table 1, Lines 214 and following, Fig. 4 and Fig. 8 below;

In the end, I suggest moderate revisions to address these points, but also to publish this article after revisions, as the data and conclusions bring some new aspects to the general discussion of coral carbonate mounds and environmental drivers.

I hope, these comments help the authors to improve their manuscript.
Kind regrads.

**Detailed remarks:**

Line 56: …develop into mounds, where coral growth and sediment infill… add a "comma" and "where", otherwise the sentence does not make sense.

Fig. 1: The color coding for the transects are not well chosen as they are hardly to be identified. Wouldn't it be better to simply use black and white including numbers? The authors could also zoom into the area as for the side view – then the transects are larger and better visible (see sketch to the right).

Line 140: space between 140 and m

Table 1: To better structure the table and compare the data, I would recommend to add columns to min, mean, max values instead of separation by comma. For downward velocities, numbers are in opposite order (max, mean, min).

Furthermore, the authors present the hydrodynamic variables for the "mound-and-corals setting" only – however, it would be helpful for the reader to compare the data with the "no-mound setting" of the smoothed seafloor – even, if they had been published earlier. For example, in Lines 176-179, the authors mention "We calculated the coral mound engineering effect by subtracting … (hydrodynamic variables) … of the simulation with smoothed bathymetry (…) from simulations with unmodified bathymetry (…)." or in the caption of Fig.6 – it would be helpful showing these data to better grasp the difference and the impact CWC reefs and mounds do have.

Line 180-181: quite confusing sentence – better rephrase.

Line 181: "hydrodynamic variables will increase if mound height increases" – this is a positive feedback mechanism – just for curiosity, do there also exist negative feedback mechanisms?
What about the situation, if mound height reaches the level of deep thermocline or water mass boundaries or change in density level – then the mounds would grow more towards the sides then further increasing the height of the mounds. In this situation, the mound height would also limit the engineering effect and influence the environmental factors supportive for the corals. It would be nice to see a discussion on this issue as well.

From Line 214 onwards you describe data of RDA Axes 3 and 4, which are not shown. I would recommend to show these data as well in Fig. 4, which could be arranged like this sketch (or put a) and B9 on top and c) and d) at the bottom):

[Figure]

Or dependent on the proportion explained by RDA axis, skip axes 4 and 5 (also in the description, as both have values below 1%) and only show axes 1–3 (also in the figure 4).

Fig. 7: in a) y-axis: add space between "Absolute" and "bigdelta"

Lines 281–283 (and in general): This conclusion may be true for the investigated Logachev mound province. However, mounds which have reached the permanent thermocline/water mass boundary/density gradient like the upper Belgica Mound chain in the Porcupine Seabight may not provide this supportive, engineering conditions to positively affect coral/mound growth. Here and elsewhere like the Norwegian reefs, the mound height may not directly affect the coral reef growth as at the (bigger) Logachev mounds. I would recommend to tone done conclusion and/or link them to the study site instead of too much generalization.

Line 326: add space between 600 and m

Line 351: replace the second "3)" with "5)"

Line 354–356: there exist earlier studies and from different disciplines showing this pattern of coral settlement and sedimentary facies on carbonate mounds, for example:

Freiwald, A., Hühnerbach, V., Lindberg, B., Wilson, J. B., and Campbell, J., 2002, The Sula Reef Complex, Norwegian Shelf: Facies, v. 47, p. 179–200.

Foubert, A., Beck, T., Wheeler, A. J., Opderbecke, J., Grehan, A., Klages, M., Thiede, J., Henriet, J.-P., and The Polarstern ARK-XIX/3A Shipboard Party, 2005, New view of the Belgica Mounds, Porcupine Seabight, NE Atlantic: preliminary results from the Polarstern ARK-XIX/3a ROV cruise, in: Freiwald, A., and Roberts, J. M. (eds.), Cold-Water Corals and Ecosystems: Springer-Verlag, Berlin, p. 403–415.

Dorschel, B., Hebbeln, D., Rüggeberg, A., And Dullo, C., 2007, Carbonate budget of a deep water coral mound: Propeller Mound, Porcupine Seabight: International Journal of Earth Sciences, v. 96, p. 73–83.

Mortensen, P. B., Hovland, M. T., Fossa, J. H., and Furevik, D. M., 2001, Distribution, abundance and size of Lophelia pertusa coral-reefs in mid-Norway in relation to seabed characteristics: Journal of the Marine Biological Association of the UK, v. 81, p. 581–597.

Wheeler, A. J., Kozachenko, M., Henry, L.-A. Foubert, A., De Haas, H., Huvenne, V. A. I., Masson, D. G., and Olu, K., 2011a, The Moira Mounds, small cold-water coral banks in the Porcupine Seabight, NE Atlantic: Part A—an early stage growth phase for future coral carbonate mounds?: Marine Geology, v. 282, p. 53–64.

Foubert, A., Huvenne, V.A.I., Wheeler, A., Kozachenko, M., Opderbecke, J., Henriet, J.P., 2011. The Moira Mounds, small cold-water coral mounds in the Porcupine Seabight, NE Atlantic: Part B - Evaluating the impact of sediment dynamics through high-re- solution ROV-borne bathymetric mapping. Mar. Geol. 282 (1–2), 65–78.

An overview:

Vertino, A., Spezzaferri, S., Rüggeberg, A., Stalder, C., Wheeler, A., and the EUROFLEETS CWC-MOIRA Cruise Scientific Party (2015) An overview on cold-water coral ecosystems and facies. Cushman Foundation Special Publication No. 44, p. 12–19.

Line 378–380 and line 384: here you should refer to mountain instead of mound. It should read: "These zones vary with relative altitude on the mountain, but not with the absolute height above the ground, because of feedbacks between the size of the mountain and the environment (…)" and "…feedbacks between the mountain and the environment, …".

Lines 388–390: I think that the authors make it a bit too simple. They should clearly differentiate between the short-term processes related to different times with long-term processes. For example, the deep winter mixing occurs during February, which correlates with the higher downward velocities (indicate in Fig. 8 the timing of the processes, especially in 8b Side view – here it looks like all processes happens at once), while the upward velocities occur during August supporting the nutrient upwelling.

The latter was also reported by Findlay et al. (2014) but only to the depths of the coral mounds. Soetart et al. (2016) then shows that the upwelling occurs also to shallow areas, but together with the February downwelling they are clearly tidally influence with strong velocities (interesting for the engineering process) during spring tides only.

In Lines 393–398 the upwelling processes are related to nutrients transported to the sea surface, where strictly speaking the reference to Findlay et al. (2014) is not supporting this. Further in the text, the authors compare that the primary productivity is stimulated by these upward water motion bringing nutrients to the surface with processes on millennial time scales (glacial-interglacial) and refer to Eisele et al. (2011), which compare coral age data with TOC mass accumulation data from a close-by ODP core offshore Mauritania indicating a possible relationship at millennial time scales, or to Wienberg et al. (2020) and (2022), which do not present any nutrient- or primary productivity-related data to compare (2020), or they compare the western Mediterranean Sea coral mound record with an eastern Mediterranean ODP core showing Monsoon-related variability of Nile river discharge (far away from the coral site) for the former study (2022). I would recommend clear separation of processes on short- and long-term and discuss it properly.

---

## Author Response (AR2)

Thursday 15 December 2023

Author's response to the second round of review of the manuscript titled "Building your own mountain: The effects, limits, and drawbacks of cold-water coral ecosystem engineering."

Dear Peter Landschützer, associate editor with Biogeosciences,

Thank you for the opportunity to resubmit our manuscript after minor revisions.

We addressed the comments from dr. Andres Rüggeberg. Our response is attached at the end of this letter. We think that we addressed all comments sufficiently and that the manuscript improved.

My apologies for the slight delay in the revisions of this manuscript.

Thank you for considering our manuscript for publication in your journal.

With kind regards,
On behalf of all co-authors,

Anna van der Kaaden

Reply to dr. Andres Rüggeberg.

General remarks:

The manuscript went already through a review process with suggested major revisions and the present revised version improved a lot. However, I still have some concerns, which I would like the authors to consider:
- I think the authors should better differentiate between short and long-term processes in their discussion, please see detailed comment for lines 388–398 in the attached pdf;
- The authors could better discuss the relationship of mound height to increasing hydrodynamic variables (see more detailed comment to Lines 181 and 281 in the attached pdf). I miss the discussion on mounds reaching a maximum height at the boundary (perm. thermocline, water mass, density) and their behavior/control at the maximum height, or mounds occurring at much shallower water depths (e.g., Norway);
- Figures and tables could be improved, see comment to Fig. 1, Table 1, Lines 214 and following, Fig. 4 and Fig. 8 in the attached pdf;

In the end, I suggest moderate revisions to address these points, but also to publish this article after revisions, as the data and conclusions bring some new aspects to the general discussion of coral carbonate mounds and environmental drivers.

I hope, these comments help the authors to improve their manuscript.
Kind regards.

Dear dr. Rüggeberg,
Thank you for your careful review and for your helpful suggestions. Indeed, we did not discuss the hypothesis of a maximum mound height and what could happen near the boundaries of water masses etc. We like the suggestion, and we are glad to include it in our discussion. Below is our reply to your detailed comments.
Kind regards,
Anna

Detailed remarks:

Line 56: …develop into mounds, where coral growth and sediment infill… add a "comma" and "where", otherwise the sentence does not make sense.

Yes, thank you.

Fig. 1: The color coding for the transects are not well chosen as they are hardly to be identified. Wouldn't it be better to simply use black and white including numbers? The authors could also zoom into the area as for the side view – then the transects are larger and better visible (see sketch to the right).

Yes, thank you. We added a zoom in on the part of the map with the transects (same as for the side view) and highlighted the transects in black, with a colour-shading. We also numbered the transects, like in the side view and labelled the different sub-figures (a, b, and

c). We also increased the size of the legend. We choose to keep the large image of the entire province since it also indicates the encircled cold-water coral mounds that are used in our analysis.

Yes, thank you.

Table 1: To better structure the table and compare the data, I would recommend to add columns to min, mean, max values instead of separation by comma. For downward velocities, numbers are in opposite order (max, mean, min).
Furthermore, the authors present the hydrodynamic variables for the "mound-and-corals setting" only – however, it would be helpful for the reader to compare the data with the "no-mound setting" of the smoothed seafloor – even, if they had been published earlier. For example, in Lines 176-179, the authors mention "We calculated the coral mound engineering effect by subtracting … (hydrodynamic variables) … of the simulation with smoothed bathymetry (…) from simulations with unmodified bathymetry (…)." or in the caption of Fig.6 – it would be helpful showing these data to better grasp the difference and the impact CWC reefs and mounds do have.

Yes, thank you. We added columns for the min, mean, and max values and changed around the values for the downward velocities. We also added a second table (Table 2) with the min, mean, and max values of the hydrodynamic variables for the unmodified and smoothed bathymetry. These are the values on the locations of the mounds (as in Fig. 1), not from the entire area. We liked the idea of adding these values. Before we simply didn't think of it.

Line 180-181: quite confusing sentence – better rephrase.

We rephrased the sentence to (Line 181):
"We define a coral mound engineering effect as a positive or negative feedback, meaning that the magnitude of the hydrodynamic variables increases or decreases resp. with increasing coral mound height. To investigate which hydrodynamic variables are influenced by the size of a coral mound we calculated…"

Line 181: "hydrodynamic variables will increase if mound height increases" – this is a positive feedback mechanism – just for curiosity, do there also exist negative feedback mechanisms?
What about the situation, if mound height reaches the level of deep thermocline or water mass boundaries or change in density level – then the mounds would grow more towards the sides then further increasing the height of the mounds. In this situation, the mound height would also limit the engineering effect and influence the environmental factors supportive for the corals. It would be nice to see a discussion on this issue as well.

Yes, there can be a negative feedback. Actually, we see that bottom current speeds are decreased at some mound sides and that the absolute effect on bottom current speeds significantly correlates with mound height, suggesting a negative feedback. To make it clearer that both feedbacks are possible, we changed the sentence on line 183: "A

significant correlation thus indicates a significant positive or negative effect of coral mound engineering on the hydrodynamic variable."

With regards to the 'maximum mound height': When performing hydrodynamic simulations around two coral mounds of increasing (and decreasing) size, we saw no such limiting effect of hydrodynamic variables on the mounds (van der Kaaden et al., 2021, Deep-Sea Res. I), even though the coral mound heights were increased up until 1.5 times their current size and through the permanent pycnocline. Of course, coral mound formation can be restrained by non-engineered factors, as we discuss in section 4.1. The engineering effects are still interesting to study, even if mounds are constrained by non-engineered processes, and the massenerhebung effect would still apply.

To the discussion on broad-scale environmental control (section 4.1), we added (line 340): "Environmental factors other than the hydrodynamic variables investigated here, might also affect coral reef growth and subsequent cold-water coral mound formation. For example, certain water masses (Schulz et al., 2020), the permanent pycnocline (White and Dorschel, 2010), internal waves (Wang et al., 2019; Wienberg et al., 2020), seawater density (Flögel et al., 2014), and (terrestrial) sediment supply (Pirlet et al., 2011; Vandorpe et al., 2017; Lo Iacono et al., 2014) have been suggested to restrain mound formation. Cold-water coral mounds can become buried following changes to the sedimentary regime (Lo Iacono et al., 2014) and for the Logachev cold-water coral mound province it has been hypothesized that the mounds stopped growing when reaching the permanent pycnocline (White and Dorschel, 2010) or the WTOW upper boundary (Schulz et al., 2020). Van der Kaaden et al. (2021) found no levelling off of the engineering effect of the coral mound on the local hydrodynamics, even when the mound was higher than at present. Such a levelling off is also not apparent in our results (Fig. 6). This underlines that, even though coral mounds engineer their local hydrodynamic environment, coral mound formation could be restricted by non-engineered environmental processes. Still, for cold-water coral mounds that are not buried it can be interesting to investigate the general hydrodynamic regime that arises around cold-water coral mounds and how this regime might explain reef zonation on mounds."

From Line 214 onwards you describe data of RDA Axes 3 and 4, which are not shown. I would recommend to show these data as well in Fig. 4, which could be arranged like this sketch (or put a) and B9 on top and c) and d) at the bottom):

Or dependent on the proportion explained by RDA axis, skip axes 4 and 5 (also in the description, as both have values below 1%) and only show axes 1–3 (also in the figure 4).

We liked your suggestion, so we added the third axis to figure 4 and removed reference to the fourth axis in the text. We added to the results (line 219): "We do not discuss the fourth axis, as it explained <1 % of the variation in benthic cover."

Fig. 7: in a) y-axis: add space between "Absolute" and "bigdelta"

Yes, thank you.

Lines 281–283 (and in general): This conclusion may be true for the investigated Logachev mound province. However, mounds which have reached the permanent thermocline/water mass boundary/density gradient like the upper Belgica Mound chain in the Porcupine Seabight may not provide this supportive, engineering conditions to positively affect coral/mound growth. Here and elsewhere like the Norwegian reefs, the mound height may not directly affect the coral reef growth as at the (bigger) Logachev mounds. I would recommend to tone done conclusion and/or link them to the study site instead of too much generalization.

We added (line 279) "Our results underline that, at the Logachev cold-water coral mound province, conditions for cold-water corals…"
We further do not claim that the mounds positively affect mound growth, but we discuss how the hydrodynamic regime that arises around coral mounds determines the facies/reef zonation on coral mounds (section 4.2). We do think that the 'hydrodynamic regime' is a general feature of coral mounds as its description is based on hydrodynamic theory of waterflow passed an object in combination with observations from several cold-water coral mounds (line 361).

Line 326: add space between 600 and m

Yes, thank you.

Line 351: replace the second "3)" with "5)"

Yes, thank you.

Line 354–356: there exist earlier studies and from different disciplines showing this pattern of coral settlement and sedimentary facies on carbonate mounds, for example:

1. Freiwald, A., Hühnerbach, V., Lindberg, B., Wilson, J. B., and Campbell, J., 2002, The Sula Reef Complex, Norwegian Shelf: Facies, v. 47, p. 179–200.
2. Foubert, A., Beck, T., Wheeler, A. J., Opderbecke, J., Grehan, A., Klages, M., Thiede, J., Henriet, J.-P., and The Polarstern ARK-XIX/3A Shipboard Party, 2005, New view of the Belgica Mounds, Porcupine Seabight, NE Atlantic: preliminary results from the Polarstern ARK-XIX/3a ROV cruise, in: Freiwald, A., and Roberts, J. M. (eds.), Cold-Water Corals and Ecosystems: Springer-Verlag, Berlin, p. 403–415.
3. Dorschel, B., Hebbeln, D., Rüggeberg, A., And Dullo, C., 2007, Carbonate budget of a deep water coral mound: Propeller Mound, Porcupine Seabight: International Journal of Earth Sciences, v. 96, p. 73–83.
4. Mortensen, P. B., Hovland, M. T., Fossa, J. H., and Furevik, D. M., 2001, Distribution, abundance and size of Lophelia pertusa coral-reefs in mid-Norway in relation to seabed characteristics: Journal of the Marine Biological Association of the UK, v. 81, p. 581–597.
5. Wheeler, A. J., Kozachenko, M., Henry, L.-A. Foubert, A., De Haas, H., Huvenne, V. A. I., Masson, D. G., and Olu, K., 2011a, The Moira Mounds, small cold-water coral banks in the Porcupine Seabight, NE Atlantic: Part A—an early stage growth phase for future coral carbonate mounds?: Marine Geology, v. 282, p. 53–64.
6. Foubert, A., Huvenne, V.A.I., Wheeler, A., Kozachenko, M., Opderbecke, J., Henriet, J.P.,

2011. The Moira Mounds, small cold-water coral mounds in the Porcupine Seabight, NE Atlantic: Part B - Evaluating the impact of sediment dynamics through high-re- solution ROV-borne bathymetric mapping. Mar. Geol. 282 (1–2), 65–78.
An overview:
Vertino, A., Spezzaferri, S., Rüggeberg, A., Stalder, C., Wheeler, A., and the EUROFLEETS CWC-MOIRA Cruise Scientific Party (2015) An overview on cold-water coral ecosystems and facies. Cushman Foundation Special Publication No. 44, p. 12–19.

Thank you for the references. Freiwald et al. (2002), Mortensen (2001), Wheeler (2011), and Foubert (2011), address the reefs that are much smaller (i.e., with a height in terms of meters, up to at most 35 m) than the coral mounds that we address (with a height of several tens of meters up to hundreds of meters), therefore we did not add these references. We added the reference to Cold-water corals and ecosystems (2005), Dorschel et al. (2005), and Vertino et al. (2014).

Line 378–380 and line 384: here you should refer to mountain instead of mound. It should read: "These zones vary with relative altitude on the mountain, but not with the absolute height above the ground, because of feedbacks between the size of the mountain and the environment (…)" and "…feedbacks between the mountain and the environment, …".

Thank you. We now wrote 'mountains' when referring to the massenerhebung effect as described for terrestrial mountains and to 'mounds' when referring to cold-water coral mounds (line 390).

Lines 388–390: I think that the authors make it a bit too simple. They should clearly differentiate between the short-term processes related to different times with long-term processes. For example, the deep winter mixing occurs during February, which correlates with the higher downward velocities (indicate in Fig. 8 the timing of the processes, especially in 8b Side view – here it looks like all processes happens at once), while the upward velocities occur during August supporting the nutrient upwelling.
The latter was also reported by Findlay et al. (2014) but only to the depths of the coral mounds. Soetart et al. (2016) then shows that the upwelling occurs also to shallow areas, but together with the February downwelling they are clearly tidally influence with strong velocities (interesting for the engineering process) during spring tides only.
In Lines 393–398 the upwelling processes are related to nutrients transported to the sea surface, where strictly speaking the reference to Findlay et al. (2014) is not supporting this. Further in the text, the authors compare that the primary productivity is stimulated by these upward water motion bringing nutrients to the surface with processes on millennial time scales (glacial-interglacial) and refer to Eisele et al. (2011), which compare coral age data with TOC mass accumulation data from a close-by ODP core offshore Mauritania indicating a possible relationship at millennial time scales, or to Wienberg et al. (2020) and (2022), which do not present any nutrient- or primary productivity-related data to compare (2020), or they compare the western Mediterranean Sea coral mound record with an eastern Mediterranean ODP core showing Monsoon-related variability of Nile river discharge (far away from the coral site) for the former study (2022). I would recommend clear separation of processes on short- and long-term and discuss it properly.

We clarified our reasoning in this paragraph and toned down our discussion on nutrient upwelling (line 407): "These nutrients might be transported by the predominantly upward water motions…". We clarified our reasoning using millennial time-scale studies (line 409): "This might benefit the coral reefs, as a global review (Maier et al., 2023) and studies on millennial time scales (Eisele et al., 2011; Wienberg et al., 2022) show the benefits of increased primary productivity for coral reef growth."

Figure 8: in panel (a) we clearly state that these processes happen in February. The engineered hydrodynamics in panel (b) happen regardless of season. In the caption for panel (b) we clarified: "The up- and downward water motions do not necessarily happen simultaneously but can result from (spring-neap) tidal motions."

In the section (4.3 Ecosystem engineering effects on cold-water corals and climate change) we explicitly discuss the ecosystem engineering effects in relation to climate change and not the non-engineered environmental processes such as deep winter mixing (discussed in section 4.1). From previous studies we know that cold-water coral mounds also influence downward water motions, but within the entire cold-water coral mound province we did not find a correlation between mound height and the absolute effect on downward velocities in February. We find it likely that this is because the signal is obscured by broad-scale environmental processes in February, e.g., deep-winter mixing. We already indicated this at the beginning of section 4.2 but now elaborated to (line 356): "Previous studies (e.g., van der Kaaden et al., 2021; Davies et al., 2009; Findlay et al., 2013; Cyr et al., 2016; Mienis et al., 2007) showed that cold-water coral mounds affect downward water motions. Our results also indicate that coral mounds influence downward velocities in February (Fig. 6). However, the magnitude of this effect does not correlate to mound height, likely because of the prevailing influence of non-engineered broad-scale environmental processes in winter (discussed in section 4.1)."